# Lionheart LincRNA alleviates cardiac systolic dysfunction under pressure overload

Yasuhide Kuwabara [1,6], Shuhei Tsuji [1,6], Masataka Nishiga [1], Masayasu Izuhara[1], Shinji Ito[2], Kazuya Nagao[3], Takahiro Horie [1], Shin Watanabe[1], Satoshi Koyama[1], Hisanori Kiryu[4], Yasuhiro Nakashima[1], Osamu Baba[1], Tetsushi Nakao[1], Tomohiro Nishino[1], Naoya Sowa[1], Yui Miyasaka[1], Takeshi Hatani[1,5], Yuya Ide[1], Fumiko Nakazeki [1], Masahiro Kimura[1], Yoshinori Yoshida [5], Tsukasa Inada[3], Takeshi Kimura[1] & Koh Ono [1✉]

Recent high-throughput approaches have revealed a vast number of transcripts with unknown functions. Many of these transcripts are long noncoding RNAs (lncRNAs), and intergenic region-derived lncRNAs are classified as long intergenic noncoding RNAs (lincRNAs). Although *Myosin heavy chain 6 (Myh6)* encoding primary contractile protein is down-regulated in stressed hearts, the underlying mechanisms are not fully clarified especially in terms of lincRNAs. Here, we screen upregulated lincRNAs in pressure overloaded hearts and identify a muscle-abundant lincRNA termed *Lionheart*. Compared with controls, deletion of the *Lionheart* in mice leads to decreased systolic function and a reduction in MYH6 protein levels following pressure overload. We reveal decreased MYH6 results from an interaction between *Lionheart* and Purine-rich element-binding protein A after pressure overload. Furthermore, human LIONHEART levels in left ventricular biopsy specimens positively correlate with cardiac systolic function. Our results demonstrate *Lionheart* plays a pivotal role in cardiac remodeling via regulation of MYH6.

[1] Department of Cardiovascular Medicine, Graduate School of Medicine, Kyoto University, Kyoto, Japan. [2] Medical Research Support Center, Graduate School of Medicine, Kyoto University, Kyoto, Japan. [3] Department of Cardiovascular Center, Osaka Red Cross Hospital, Osaka, Japan. [4] Department of Computational Biology and Medical Sciences, Graduate School of Frontier Sciences, University of Tokyo, Chiba, Japan. [5] Center for iPS Cell Research and Application, Kyoto University, Kyoto, Japan. [6] These authors contributed equally: Yasuhide Kuwabara, Shuhei Tsuji. ✉email: kohono@kuhp.kyoto-u.ac.jp

Heart failure caused by cardiac dysfunction is a refractory condition, in which the heart cannot pump the blood to each organ sufficiently. The insufficient pumping ability leads to symptoms such as dyspnea and edema. Despite recent treatment advances, heart failure is a significant burden worldwide. Thus, identifications of novel therapeutic targets are urgently needed[1]. Recent genome-wide approaches using next-generation sequencing have revealed a vast number of transcripts with unknown functions within the human genome[2–4]. Many of these transcripts are long noncoding RNAs (lncRNAs), and lncRNAs derived from intergenic regions between two protein coding genes are classified as long intergenic noncoding RNAs (lincRNAs)[3]. At present, the functions of lincRNAs in the context of heart failure are not fully defined; therefore, exploring the molecular functions of lincRNAs in heart failure could lead to novel therapeutic strategies.

Myosin heavy chain proteins or myosin heavy polypeptides (MYH), which exist in two isoforms in the heart, MYH6 and MYH7, are abundantly expressed sarcomere proteins. Because the rate of ATP hydrolysis by MYH6 is three times higher than by MYH7, MYH6 has been associated with fast contractility, whereas MYH7 is linked to slower contractility. The expression levels of MYH6 have been reported to decrease during ageing and in disease states[5], and there has been great interest in revealing the molecular mechanisms by which MYH6 expression is regulated during heart failure[6].

Purine-rich element-binding protein A (PURA) is a known DNA- and RNA-binding protein that regulates DNA replication, transcription, and translation through binding to purine-rich elements within the PURA targets[7]. PURA is expressed ubiquitously, and PURA-deficient mice die shortly after birth due to neurological and hematopoietic abnormalities[8]. In cardiomyocytes, PURA binds to a purine-rich negative regulatory (PNR) element in the first intron of the *Myh6* locus leading to decreased expression of *Myh6*[9,10]. PURA also binds to Myh6 mRNA resulting in decreased translation of the transcript[10]. However, it remains unclear how PURA functions are regulated in cardiac remodeling.

Here, we screen intergenic regions of the genome for lincRNAs whose expressions increase during cardiac remodeling. We identify Lionheart as a muscle-abundant lincRNA and generate *Lionheart* knockout (*Lionheart*-KO) mice, which have decreased cardiac systolic function and reduced *Myh6* expression in pressure overload conditions relative to controls. Furthermore, we determine that the downregulation of *Myh6* results from the loss of an interaction between Lionheart and PURA under pressure overload. Our results reveal that the lincRNA, Lionheart, plays pivotal roles in cardiac remodeling by regulating the *Myh6* expression.

## Results

### Identification of lincRNAs upregulated by pressure overload.
To identify transcribed intergenic regions that yield upregulated lincRNAs during cardiac remodeling, we performed a screen using total RNA extracted from mouse hearts subjected to transverse aortic constriction (TAC) surgery. In this screen, we assessed two time points at 2 and 8 weeks after TAC surgery for the hypertrophic response and the heart failure phase, respectively. Focusing on the intergenic regions that were yielding lincRNAs with higher expression levels after TAC than sham controls, our microarray screen identified 10 intergenic regions and we numbered the transcripts lincRNA-1 to -10 (Fig. 1a). We validated the upregulated expression levels of lincRNA-3, -5, and -7 at 2 and 8 weeks after TAC surgery by quantitative PCR (Fig. 1b, c). Among three lincRNAs, we focused on lincRNA-5,

because the DNA sequence of this locus was the most highly conserved in mammals (Supplementary Fig. 1). We termed the lincRNA-5 as long intergenic-origin noncoding RNA in heart or Lionheart. To further confirm the upregulation of Lionheart by hypertrophic stimuli, we examined the Lionheart levels in hypertrophied mouse hearts induced by phenylephrine and iso-proterenol administration and in neonatal mouse cardiomyocytes (NMCMs) treated with angiotensin II or phenylephrine. Expression levels of *Lionheart* were increased in both models (Fig. 1d, e). Rapid amplification of cDNA ends (RACE) analyses revealed that Lionheart was 402 nucleotides long, capped, spliced, and polyadenylated (Fig. 1f, g, and Supplementary Fig. 2a). Although Lionheart is currently annotated as BY787644 and on the minus strand of Gm13943, Lionheart is longer than these annotated transcripts and the DNA position was mm10_chr2:77,314,725–77,317,068.

### Features and conservation of Lionheart.
We determined the organ distribution of Lionheart in adult mice and found that Lionheart is abundantly expressed in striated muscle. To assess Lionheart level in cardiac myocytes, we sorted NMCMs and cardiac fibroblasts, and revealed that Lionheart levels were higher in NMCMs than that in cardiac fibroblasts (Fig. 2b). Subcellular fractionation experiments demonstrated that the nuclear/cytoplasmic ratio pattern of Lionheart was quite similar to other, nuclear transcripts including U2 snRNA and Xist (Fig. 2c). These data indicate that Lionheart predominantly exists in the nucleus of cardiac myocytes. Furthermore, PhyloCSF bioinformatic analysis[11] and in vitro translation assays confirmed that Lionheart is a noncoding RNA (Supplementary Fig. 2b–d). We next examined the degree of *Lionheart* conservation in mammals. Because the first exon of mouse *Lionheart* is well conserved in mammals (Supplementary Fig. 1) and the neighboring genes of mouse *Lionheart* are identical in rat and human (Supplementary Fig. 3), we designed specific primers for rat and human orthologous sites for the first exon of mouse *Lionheart* and evaluated the levels of Lionheart in rats and humans. Lionheart level in neonatal rat cardiac myocytes was significantly higher than in cardiac fibroblasts (Mann–Whitney test, $p = 0.0286$; Fig. 2d). We generated human induced pluripotent stem cells (hiPSCs)-derived cardiomyocytes and demonstrated human LIONHEART level was higher in hiPSCs-derived cardiomyocytes than that in undifferentiated hiPSCs (Fig. 2e).

### Regulation of Lionheart promoter activity.
To reveal how *Lionheart* expression is regulated, we performed a promoter assay and found that a transcription factor, serum response factor (SRF), significantly increased the activity of a 2.0 kilobase (kb) *Lionheart* promoter (one-way ANOVA, $p < 0.0001$; Fig. 2f). Chromatin immunoprecipitation (ChIP)-sequencing also demonstrated SRF and P300 binding to the *Lionheart* promoter (Supplementary Fig. 4a, b). Because SRF-mediated reporter activities were similar between the −0.6-kb and the −2.0-kb promoter construct, we focused on the −0.6 kb *Lionheart* promoter and tried to identify SRF binding sites. To this end, we searched for SRF consensus binding sequences, CArG boxes[12], and identified six putative CArG boxes located within the −0.6 kb promoter (Supplementary Fig. 4c). We confirmed that SRF bound at least two CArG boxes located within the promoter region of *Lionheart* using site-directed mutagenesis (Fig. 2g and Supplementary Fig. 5).

### Generation of Lionheart knockout mice and the phenotype.
To explore the functions of Lionheart in vivo, we generated *Lionheart* knockout (*Lionheart*-KO) mice (Supplementary Fig. 6) and performed TAC surgery in *Lionheart*-KO mice. Lionheart levels

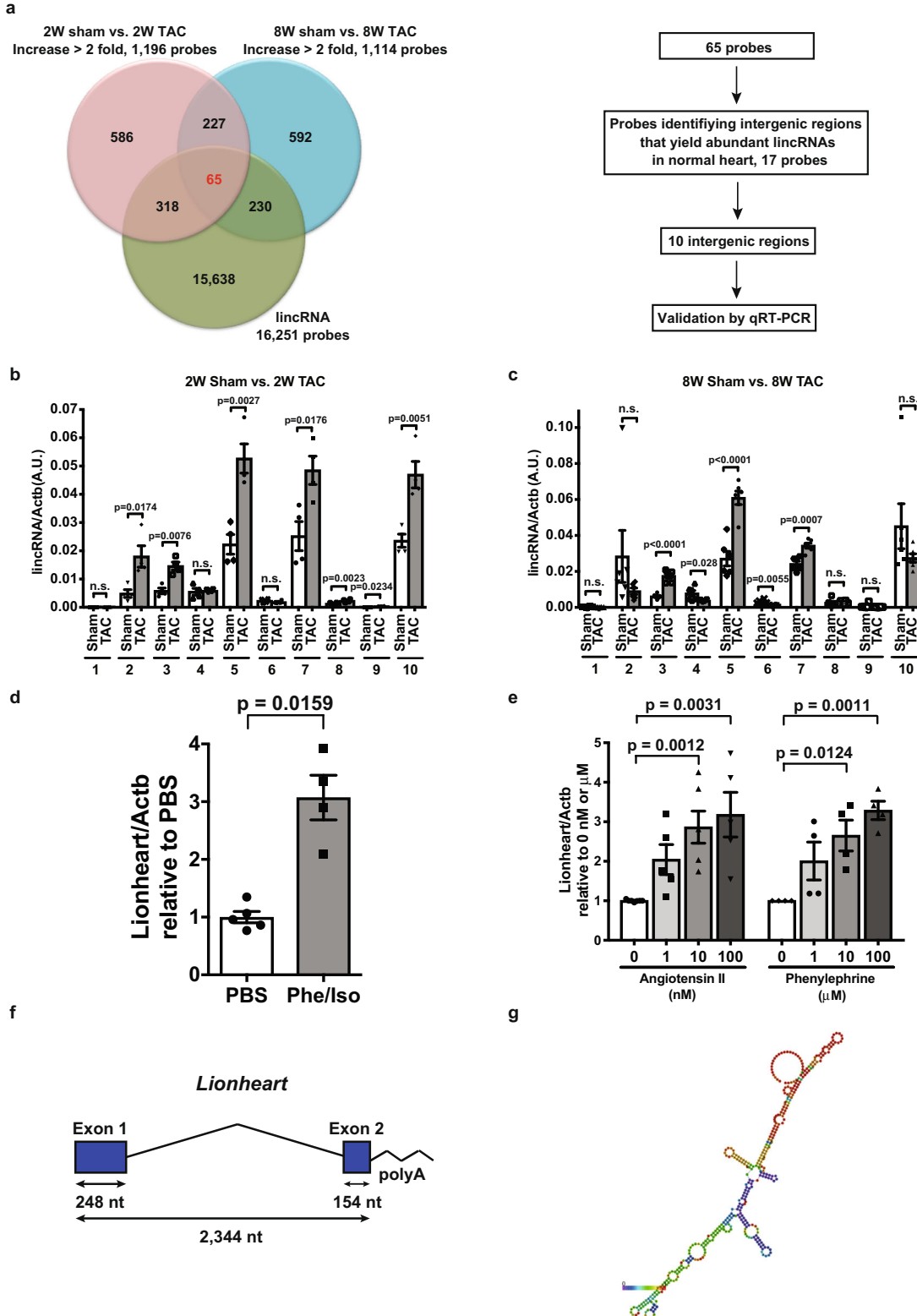

**Fig. 1 Identification of Lionheart and the structure of this lincRNA. a** Screening of intergenic regions that yielded lincRNAs during cardiac remodeling. Venn diagram shows the results of the microarray analyses. Each number represent the number of probes applied. Among 65 probes, we focused on probes that identified intergenic regions that were transcribed actively in normal adult mouse hearts. We tested the validation of the 10 lincRNAs. 2 W: 2 weeks; 8 W: 8 weeks; TAC: transverse aortic constriction. **b** Validation at 2 weeks after TAC. *n* = 4 in all groups. **c** Validation at 8 weeks after TAC. *n* = 6 in all groups. **d** Lionheart levels in the hearts of mice that were administered phenylephrine (Phe) and isoproterenol (Iso). PBS: phosphate-buffered saline. PBS group: *n* = 5; Phe/Iso group: *n* = 4. **e** Lionheart levels in neonatal mouse cardiomyocytes (CMs) stimulated with angiotensin II (0 nM: *n* = 7; 1 nM: *n* = 5; 10 nM: *n* = 6; 100 nM: *n* = 5) or phenylephrine (*n* = 4 in all groups). **f** Structure of *Lionheart*. **g** Predicted secondary structure of Lionheart.

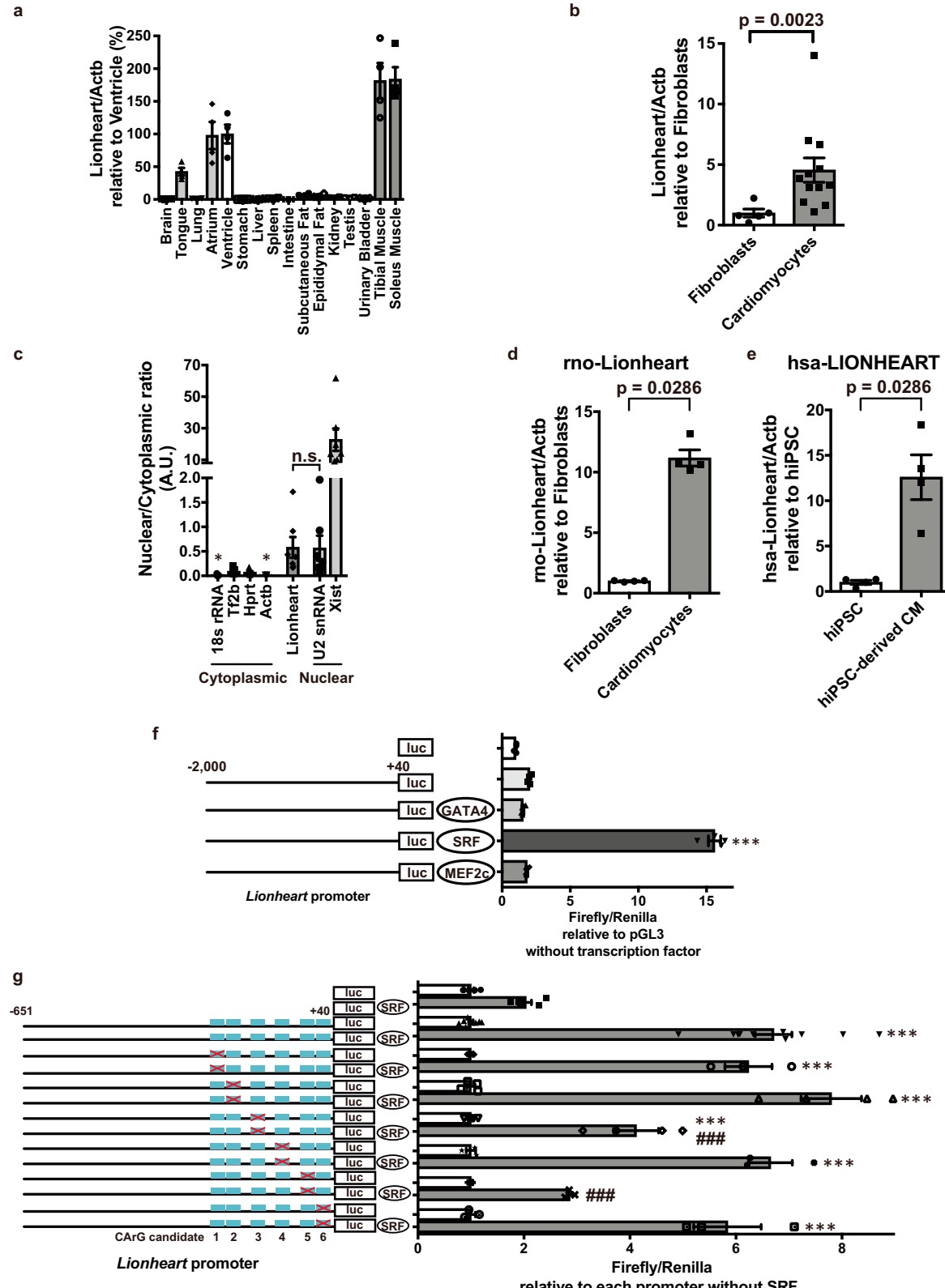

were increased following TAC surgery in control hearts but not in *Lionheart*-KO hearts (Fig. 3a). Although there was no difference in heart weight (HW)/body weight (BW) ratios between control and *Lionheart*-KO mice at 8 weeks after TAC surgery (Supplementary Fig. 7a, b), echocardiography demonstrated that end-diastolic interventricular septum thickness was thinner in *Lionheart*-KO mice than in control mice (Fig. 3b, c). We observed

a reduction in the cross-sectional area of cardiomyocytes in *Lionheart*-KO hearts compared with controls post TAC surgery (Fig. 3d, e). Systolic cardiac function declined in *Lionheart*-KO hearts at 5 and 8 weeks post TAC surgery (Fig. 3f and Supplementary Fig. 7c). In line with the reduced cardiac function, *Nppb* mRNA levels in *Lionheart*-KO hearts were higher than in control hearts after TAC (Fig. 3g). Intriguingly, we detected a significant

**Fig. 2 Features of mouse Lionheart and the conservation in rat and human. a** Tissue distribution of Lionheart in adult mouse. $n = 4$ in all organs.
**b** Relative expression of Lionheart in neonatal mouse cardiomyocytes (CMs) ($n = 12$) compared with cardiac fibroblasts ($n = 5$). **c** Nuclear and cytoplasmic distribution of several transcripts in adult mouse hearts. $n = 7$ in all transcripts. *, compared with Lionheart. **d** Lionheart levels in neonatal rat cardiac fibroblasts and CMs. $n = 4$ in both groups. **e** LIONHEART levels in human iPSCs and human iPSCs-derived CMs. $n = 4$ in both groups. **f** $-2.0$ kb *Lionheart* promoter activities with transcription factors (TFs). $n = 4$ in all groups. ***, compared with empty promotor vector without TFs. **g** Promoter activities with mutations in possible CArG boxes. Crosses in red indicate mutated possible CArG boxes. Empty vector: $n = 6$; $-0.6$ kb intact *Lionheart* promoter: $n = 10$; Mutant 1: $n = 3$; Mutants 2 and 3: $n = 4$; Mutants 4, 5, and 6: $n = 3$. ***, compared with empty vector with SRF; ###, compared with $-0.6$ kb intact *Lionheart* promotor with SRF.

reduction in *Myh6* transcript levels in *Lionheart*-KO hearts compared with wild-type hearts after TAC (one-way ANOVA, $p < 0.0041$; Fig. 3h). The protein levels of MYH6 in *Lionheart*-KO hearts were also reduced compared with wild-type hearts at 8 weeks after TAC (Fig. 3i, j). The transcript and protein levels of *Myh7* were unchanged between wild-type and *Lionheart*-KO hearts (Supplementary Fig. 7d–g). Compared with control hearts, cardiac fibrosis was attenuated in *Lionheart*-KO hearts after TAC (Supplementary Fig. 7h–k). Because blood pressure also affects cardiac hypertrophy and fibrosis, we measured blood pressure in control and *Lionheart*-KO mice. However, there was no difference in blood pressure between the two groups (Supplementary Fig. 8).

**Revealing the molecular function of Lionheart.** It has been reported that lncRNAs can act in either *cis* or *trans*[13]. To assess whether Lionheart works in *cis*, we evaluated the expression levels of *Lionheart*-flanking genes, namely *Sestd1* and *Zfp385b*. As shown in Supplementary Fig. 9, there was no difference in the expression levels of *Sestd1* and *Zfp385b* between control and *Lionheart*-KO mice. These data strongly suggested that Lionheart does not work in *cis*.

To assess the global expression changes of other previously identified RNAs in *Lionheart*-KO mouse hearts after TAC surgery, we performed microarray analysis for the levels of primary-microRNAs (pri-miRNAs) and other lncRNAs. The levels of pri-miRNA-669a-3, pri-miRNA-709, pri-miRNA-1946b, and pri-miRNA-3962 were more than two times higher in *Lionheart*-KO mice than control mice (Supplementary Fig. 10a). Concerning the other lncRNAs, 5 lncRNAs were increased (fold change > 2) and 2 lncRNAs were decreased (fold change < 0.5) in *Lionheart-KO* mice compared with control mice (Supplementary Fig. 10b). Furthermore, we conducted qPCR and evaluated the expression levels of previously identified lncRNAs that play pivotal roles in cardiac remodeling with pressure overload such as myosin heavy chain-associated RNA transcripts (Mhrt)[14], cardiac hypertrophy-associated transcript (Chast)[15], cardiac hypertrophy-associated epigenetic regulator (Chaer)[16], and cardiac hypertrophy-related factor (CHRF)[17]. As shown in Supplementary Fig. 10c–f, there were no significant differences in these lncRNA levels between control and *Lionheart*-KO mice (one-way ANOVA).

We next sought to identify Lionheart-binding proteins in nuclear extracts of hearts following TAC surgery. Using a Lionheart-antisense-transcript as a control, RNA pull-down assays followed by mass spectrometry identified 46 Lionheart-specific-binding candidates (sense/antisense: more than 1.5-fold, Fig. 4a). Focusing on these protein candidates, Gene Ontology-Biological Process (GO-BP) enrichment terms were mitochondrial functions, metabolic processes, and heart contraction (Supplementary Table 1). When we focused on the antisense-specific candidates (antisense/sense: more than 1.5-fold), GO-BP enrichment terms were extracellular matrix organization, collagen-associated signaling pathway, and angiogenesis (Supplementary Table 2). We think that these unbiased analyses suggest

that Lionheart might be involved in mitochondrial functions, metabolic processes, and heart contraction. Hence, we evaluated mitochondrial structure by transmission electron microscopy. However, there was no obvious difference in mitochondrial structures and sizes between control and *Lionheart*-KO mouse hearts (Supplementary Fig. 11).

Of the Lionheart-specific-binding protein candidates, we further focused on PURA, because our unbiased RNA pull-down assays demonstrated that PURA interacted with Lionheart with the highest specificity (Fig. 4a). Independent RNA pull-down assays confirmed that Lionheart bound to PURA in heart nuclear extracts (Fig. 4b). RNA immunoprecipitation using heart nuclear proteins and an anti-PURA antibody also demonstrated that PURA bound to Lionheart in the nucleus and the binding was increased after TAC (Fig. 4c, d), whereas pressure overload did not affect the PURA protein levels (Supplementary Fig. 12). To determine if the observed Lionheart upregulation is a left ventricle-specific change in hearts with TAC or not, we evaluated Lionheart levels in right ventricles of the heart after TAC and found that Lionheart upregulation was not observed in right ventricles after TAC (Supplementary Fig. 13a). Western blotting for PURA also demonstrated that PURA levels did not change in right ventricles after TAC (Supplementary Fig. 13b, c). These data indicate that TAC surgery resulted in left ventricle-specific Lionheart upregulation. In addition, we sought to characterize the Lionheart–PURA interaction at the molecular level. It was reported that PURA can bind to purine-rich elements in DNA or RNA[7,18]. Bioinformatic analyses demonstrated that nucleotides (nts) 60–100 of Lionheart are purine-rich and predicted to have the highest binding capacity for PURA (Fig. 4e–h). We constructed several shortened Lionheart transcripts with or without nts 60–100 and carried out RNA pulldown using these transcripts and nuclear extracts from TAC hearts. Our data demonstrate that the nts 60–100 of Lionheart are necessary for binding to PURA (Fig. 4i–k).

Previous reports indicated that PURA negatively regulates *Myh6* transcription by binding to a PNR element in the first intron of the *Myh6* gene[9,10]. Our data demonstrated that PURA bound to Lionheart, and that *Myh6* transcript levels were decreased in the hearts of *Lionheart*-KO mice after TAC surgery compared with controls. Thus, we hypothesized that the absence of interaction between PURA and Lionheart in *Lionheart*-KO hearts may lead to further PURA binding to the PNR element of *Myh6* locus resulting in decreased *Myh6* transcript levels after TAC. To test this hypothesis, we performed ChIP experiments using an anti-PURA antibody and nuclear extracts from the hearts of mice subjected to TAC. We observed that the PNR element was enriched in *Lionheart*-KO hearts compared with control hearts after TAC (Fig. 4l). To demonstrate the direct competitive function of Lionheart in binding between PURA and the PNR element, we performed RNA electrophoretic mobility shift assay (EMSA) and demonstrated that Lionheart bound to PURA with higher affinity than the binding capacity of PURA to the PNR element (Fig. 4m, n). Furthermore, Lionheart overexpression-mediated *Myh6* promoter activity was decreased

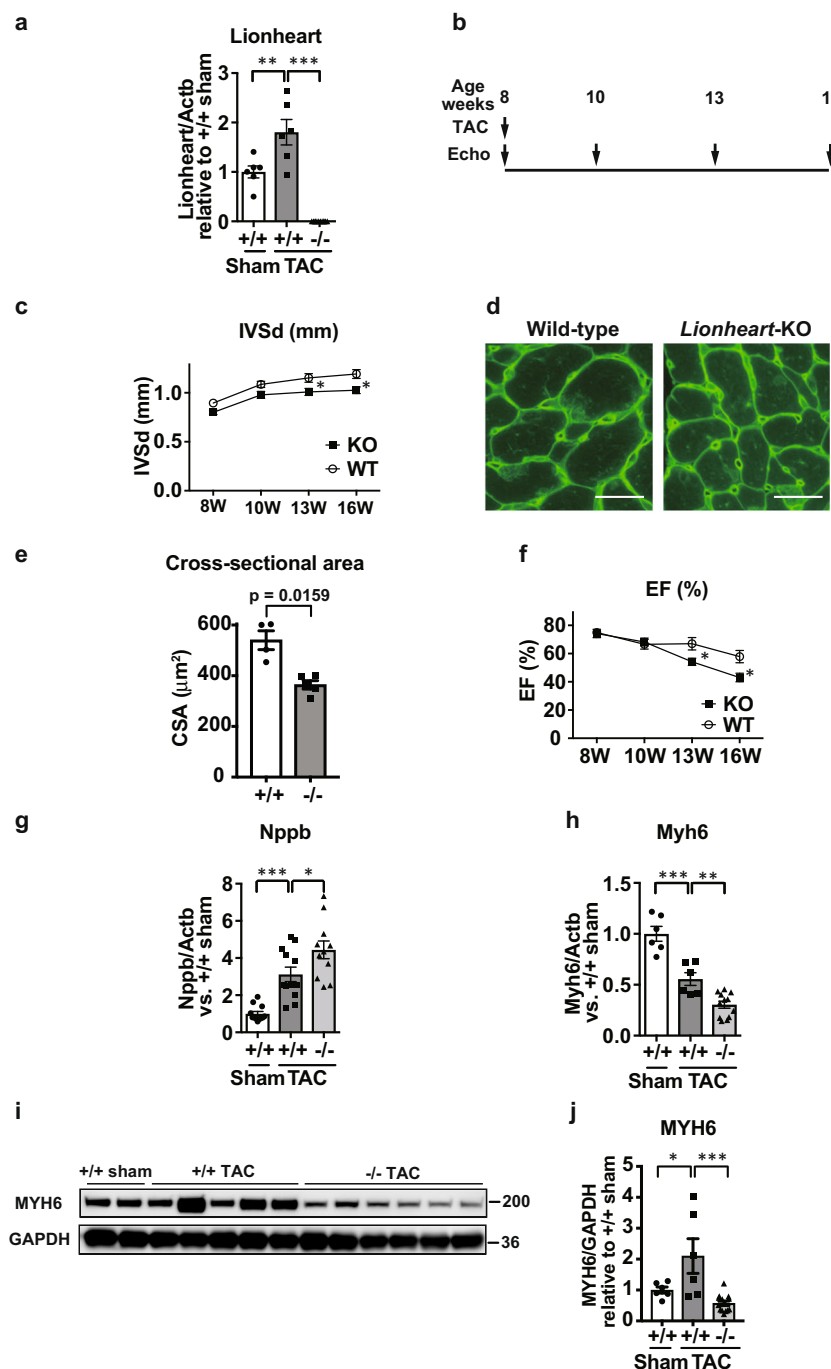

**Fig. 3 Analyses of *Lionheart* knockout mice subjected to pressure overload. a** Lionheart levels after TAC. +/+ sham and +/+ TAC groups: $n = 6$; −/− TAC group: $n = 12$. **b** Assessment of cardiac phenotype after TAC by echocardiography. **c** Changes in interventricular septal thickness at end-diastole (IVSd). 8 W: 8 weeks; 10 W: 10 weeks; 13 W: 13 weeks; 16 W: 16 weeks. WT 8 W and WT 10 W: $n = 12$; WT 13 W: $n = 10$; WT 16 W: $n = 8$; KO 8 W: $n = 12$; KO 10 W, KO 13 W, and KO 16 W: $n = 17$. *, compared with WT at same age. **d** WGA staining after TAC. Scale bar = 20 μm. **e** Quantification of the cross-sectional area. +/+ group: $n = 4$; −/− group: $n = 5$. **f** Changes in EF. WT 8 W and WT 10 W: $n = 14$; WT 13 W: $n = 12$; WT 16 W: $n = 8$; KO 8 W: $n = 12$; KO 10 W, KO 13 W, and KO 16 W: $n = 17$. *, compared with WT at same age. **g** mRNA levels of Nppb. +/+ sham and +/+ TAC groups: $n = 12$; −/− TAC group: $n = 11$. **h** mRNA levels of Myh6. +/+ sham and +/+ TAC groups: $n = 6$; −/− TAC group: $n = 12$. **i, j** MYH6 protein levels in *Lionheart*-KO mouse hearts after TAC. **i** Representative images of western blotting for MYH6 and GAPDH, as a loading control. **j** Quantification of the western blotting data. +/+ sham and +/+ TAC groups: $n = 6$; −/− TAC group: $n = 14$.

when the PNR element was deleted in the *Myh6* promoter (Fig. 4o). To determine if *Myh6* is a downstream target of Lionheart, we overexpressed Lionheart in NMCMs and found that Lionheart overexpression resulted in *Myh6* upregulation at the transcript and protein levels (Fig. 4p–r). These data indicate that Lionheart acts in *trans* and regulates MYH6 level

by inhibiting PURA binding to the PNR element in the *Myh6* locus.

**Rescue of Lionheart-KO heart phenotype by AAV9-Lionheart.**
Based on the data above, Lionheart functions in *trans*. However, we could not exclude a possibility that the observed phenotype in

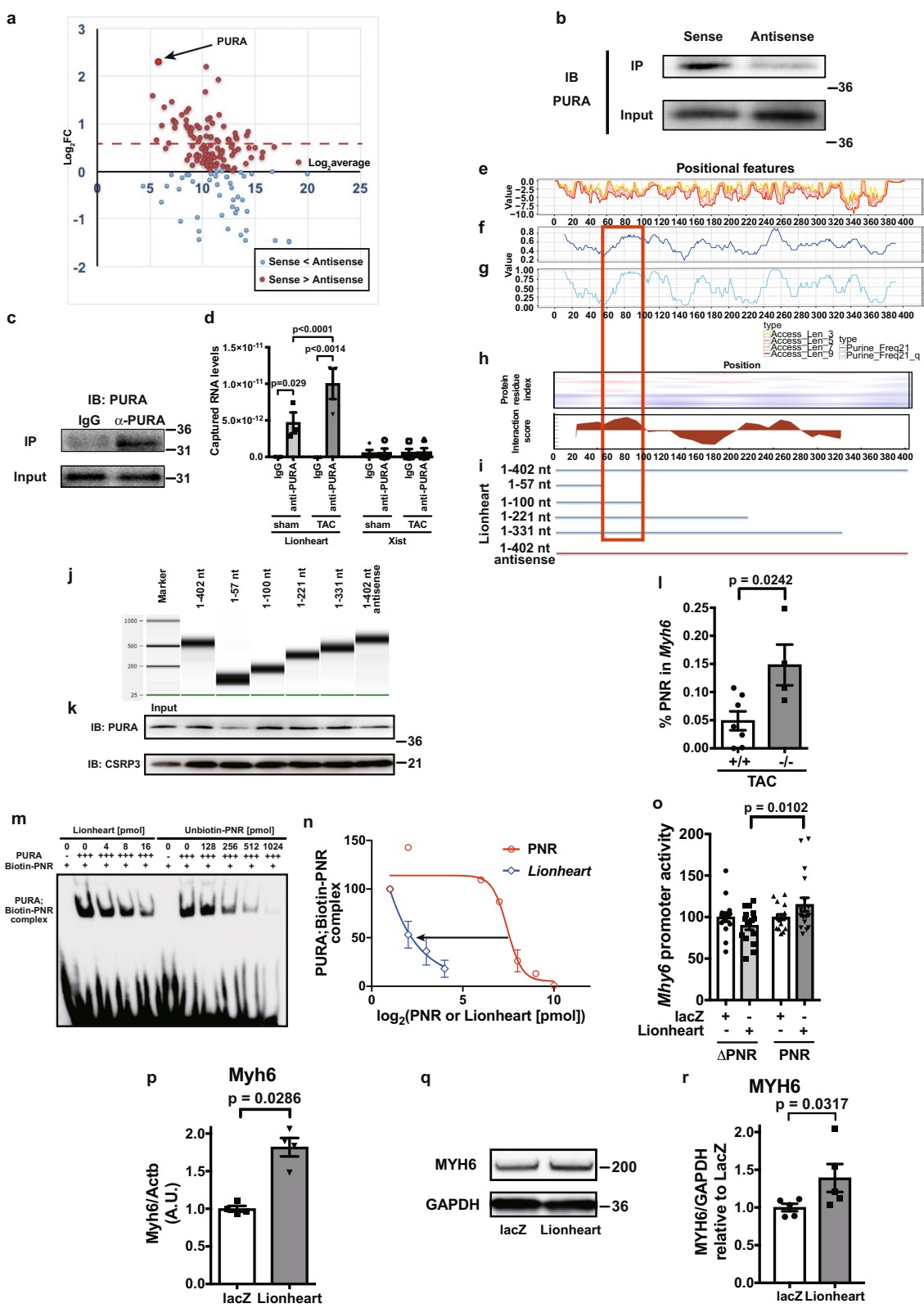

*Lionheart*-KO mice resulted from the DNA deletions of *Lionheart* loci in the genome[19,20]. To exclude this possibility and to confirm that the Lionheart transcript is essential, we sought to rescue the phenotype in *Lionheart*-KO heart with Adeno-associated virus 9 (AAV9) harboring *Lionheart*. We injected AAV9-Control or AAV9-*Lionheart* in *Lionheart*-KO mice at P3 and performed TAC surgery at 8 weeks old. At 8 weeks post TAC, we assessed

cardiac systolic function by echocardiography (Fig. 5a). Lionheart expression was restored (Fig. 5b) and cardiac systolic function was improved by AAV9-*Lionheart* injection compared with AAV9-Control injection (Fig. 5c). We also evaluated the *Myh6* mRNA and protein levels in the hearts of AAV9-Control-injected mice and AAV9-*Lionheart*-injected mice. Both mRNA and protein levels of *Myh6* were upregulated in hearts of

**Fig. 4 Identification of Lionheart-binding protein and the molecular function of the interaction. a** Result of mass spectrometry to identify Lionheart-binding proteins. All isolated proteins by RNA pulldown were analyzed. Horizontal and vertical axes indicate the abundance and the specificity of each protein, respectively. The dotted line in red indicates the 1.5-fold change boundary. **b** Western blotting for PURA isolated by RNA pulldown. IB: immunoblot. **c** Confirmation of PURA precipitation with an anti-PURA antibody in RNA immunoprecipitation. **d** RNA immunoprecipitation using an anti-PURA antibody. Lionheart sham IgG: $n = 4$; Lionheart sham anti-PURA, Lionheart TAC IgG, and Lionheart TAC anti-PURA: $n = 3$; Xist sham IgG, Xist sham anti-PURA, Xist TAC IgG, and Xist TAC anti-PURA: $n = 4$. **e–h** Bioinformatic analyses for structural features of Lionheart and the interaction between Lionheart and PURA. **e** Accessibility of each nucleotide in Lionheart. **f, g** The percentage of purine frequency in each 21-nucleotide section (**f**) and quantile value (**g**). **h** catRAPID analyses. **i** The Lionheart fragmentation used in the RNA pulldown, as shown in (**j**) and (**k**). **j** Confirmation of each truncated Lionheart length using Bioanalyzer. **k** Western blotting for PURA and CSRP3 precipitated by RNA pulldown. **l** Enrichment of the PNR element in the *Myh6* locus by ChIP using an anti-PURA antibody. $+/+$ TAC: $n = 7$; $-/-$ TAC: $n = 4$. **m, n** Results of RNA electrophoretic mobility shift assay (EMSA). **m** Representative image of RNA EMSA. **n** Quantification of RNA-EMSA data. PNR: $n = 1$–$2$; Lionheart $n = 2$. **o** *Myh6* promoter activities activated with Lionheart in neonatal mouse cardiac myocytes (NMCMs). ΔPNR indicates the *Myh6* promoter in which PNR element was deleted. ΔPNR: $n = 16$; PNR: $n = 18$. **p** Myh6 mRNA level in NMCMs with Lionheart overexpression. $n = 4$ in both groups. **q, r** MYH6 protein level in NMCMs with Lionheart overexpression. **q** Representative images of western blotting for MYH6 and GAPDH, as a loading control. **r** Quantification of the western blotting data. $n = 5$ in both groups.

---

AAV9-*Lionheart* injected mice compared with controls (Fig. 5d–f). These data strongly suggest that the Lionheart transcript regulated MYH6 levels and improved cardiac systolic function after pressure overload in *Lionheart*-KO mice.

**Association study with human heart biopsy specimens.** Because LIONHEART was detectable in hiPSCs-derived cardiomyocytes (Fig. 2e) and *Lionheart* knockout in mice resulted in exacerbation of cardiac systolic function following TAC, we hypothesized that hsa-LIONHEART levels in human heart samples may be associated with cardiac systolic function and hemodynamic parameters. To test this hypothesis, we measured the expression levels of LION-HEART in human left ventricular biopsy specimens and evaluated the correlation between the LIONHEART expression levels and ejection fraction (EF), hemodynamic parameters, specimen *NPPB* mRNA levels, and serum BNP levels. Patient characteristics were listed in Supplementary Table 3. The expression of LIONHEART showed a positive correlation with EF and cardiac index (Fig. 6a, b). Conversely, LIONHEART expression negatively correlated with pulmonary capillary wedge pressure, specimen *NPPB* mRNA levels, and serum BNP levels (Fig. 6c–e). These data suggest that reduced LIONHEART levels in human hearts are associated with reduced cardiac systolic function.

## Discussion

In the present study, we identified a functionally uncharacterized lincRNA, Lionheart, that is expressed abundantly in striated muscle and is upregulated in the heart during pathological cardiac remodeling. Lionheart expression is driven via its own promoter activated by a transcription factor, SRF. The *Lionheart*-KO mice study revealed that Lionheart is required to prevent the exacerbation of cardiac systolic function with pressure overload. Furthermore, *Myh6* mRNA and protein levels were decreased in these mice compared with controls. Mechanistically, we demonstrated that Lionheart interacted with PURA and this interaction prevented PURA from binding to the PNR element at the *Myh6* locus. Finally, we showed that LIONHEART was detectable in human left ventricle specimens and the LIONHEART levels were positively associated with cardiac systolic function.

High-throughput analyses revealed that many lncRNAs with unknown function are deregulated in cardiac hypertrophy and failure[21], and several reports demonstrated that lncRNAs act with binding partners and regulate the expression of other cardiac proteins. Focusing on lncRNAs abundantly expressed in cardiomyocytes, four lncRNAs have been reported to be involved in cardiac remodeling in vivo[14–17,22]. These lincRNAs are Mhrt[14], Chast[15], Chaer[16], and CHRF[17]. Mhrt is the first example of lncRNA involved in cardiac remodeling and is protective in

pressure-overloaded conditions by antagonization of Brahma-related gene 1 (Brg1), a chromatin-remodeling factor[14]. Chast is an upregulated lncRNA in TAC hearts and is an inducer of cardiac hypertrophy by negatively regulating pleckstrin homology domain-containing protein family M member 1 (*Plekhm1*)[15]. Chaer interacts with the catalytic subunit of polycomb repressor complex 2 and induces greater cardiac hypertrophy after TAC[16]. CHRF was identified as a competing endogenous RNA of miRNA-489, which negatively regulates myeloid differentiation primary response gene 88 (Myd88)[17]. Because MYD88 induces cardiac hypertrophy, CHRF regulates cardiac remodeling. In addition to these four lncRNAs, our study demonstrated that Lionheart also regulates cardiac remodeling through the regulation of MYH6 expression. Considering these findings, human diseases are caused by extremely complex machineries consisting of proteins, microRNAs, and lncRNAs.

To assess if *Lionheart* knockout in mice leads to deregulation of miRNAs and other lncRNAs, we carried out microarray analysis and demonstrated that several pri-miRNAs and lncRNAs are deregulated in *Lionheart*-KO mouse hearts (Supplementary Fig. 10a, b). Among the upregulated miRNAs, it is reported that long-term miRNA-669a overexpression with AAV9 ameliorates cardiac systolic dysfunction in a dystrophic mouse model[23]. Thus, we are speculating that the miRNA-669a upregulation in *Lionheart*-KO mouse hearts is a compensatory machinery against cardiac systolic dysfunction observed in *Lionheart*-KO mice with pressure overload. Among the deregulated 7 lncRNAs, a paper showed that Snora75, a noncoding small nucleolar RNA, is negatively regulated by miRNA-24 which is enriched in platelet-derived microparticles (PMPs). While the PMPs treatment led to tumor cell apoptosis resulting in inhibition of cancer cell growth in vivo and in vitro[24], the functions of Snora75 remain to be clarified and further investigations are needed.

Many lncRNAs are presumed to regulate the expression of either their flanking genes in *cis* or distant genes in *trans*. To examine whether Lionheart acts in *cis*, we assessed the expression changes of neighboring genes, *Sestd1* and *Zfp385b*, in *Lionheart*-KO mice. Although there was no difference in *Sestd1* mRNA expression, *Zfp385b* mRNA expression tended to be reduced in *Lionheart*-KO mouse hearts both at baseline and after TAC compared with controls (Supplementary Fig. 9). Because the functions of ZFP385b, a zinc finger protein family member, have not been explored, ZFP385b might be contributory to the phenotype observed in *Lionheart*-KO mice. However, the reduction of Zfp385b in *Lionheart*-KO mice was very modest compared with the reduction of *Myh6* in *Lionheart*-KO mice. Another possibility for Lionheart function was that the act of transcription per se through the *Lionheart* locus could be crucial in the phenotype of *Lionheart*-KO mice because our loss-of-function

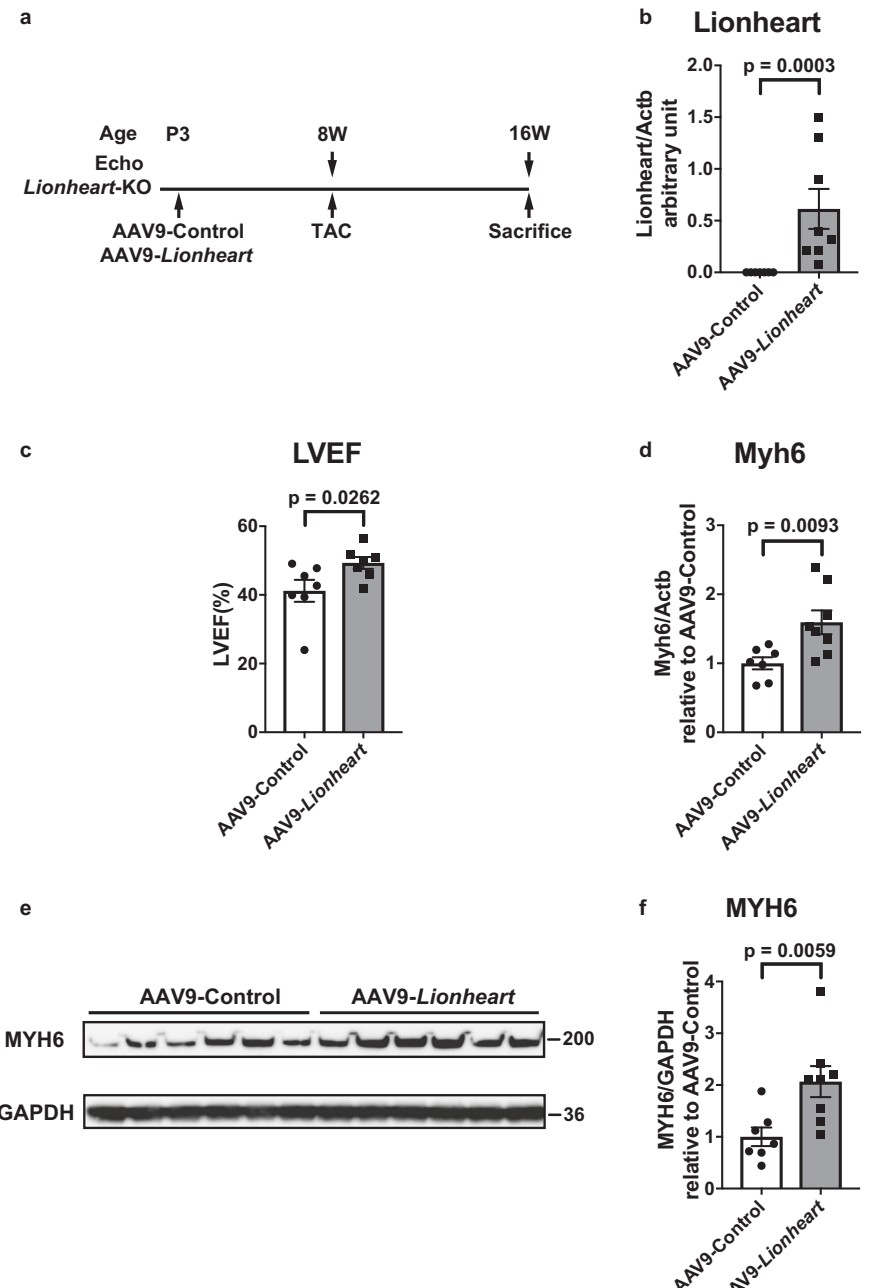

**Fig. 5 Rescue experiment of the phenotype observed in *Lionheart*-KO mouse hearts using AAV9-*Lionheart*. a** Assessment of cardiac phenotype in AAV9-injected mice subjected to TAC by echocardiography. **b** Lionheart levels at 16 weeks old. AAV9-Control group: $n = 7$; AAV9-*Lionheart* group: $n = 8$. **c** LVEF in AAV9-Control or AAV9-*Lionheart* injected mouse hearts at 16 weeks old. $n = 7$ in both groups. **d** Myh6 mRNA levels. AAV9-Control group: $n = 7$; AAV9-*Lionheart* group: $n = 8$. **e** Representative images of western blotting for MYH6 and GAPDH, a loading control. **f** Quantification of the western blotting for MYH6. AAV9-Control group: $n = 7$; AAV9-*Lionheart* group: $n = 8$.

strategy in vivo was *Lionheart* locus deletion by which the DNA element was removed completely from the genome[19,20]. Thus, the transcription of *Lionheart* locus was eliminated in *Lionheart*-KO mice. However, this possibility was also excluded by rescue experiments with AAV9-*Lionheart*. Because Lionheart transcripts derived from independent transgenes rescued the phenotype of *Lionheart*-KO mice, we concluded that Lionheart acts bona fide in *trans*.

Our data obtained from experiments using hiPSCs-derived CMs (Fig. 2e) and human ventricular biopsy specimens demonstrated that LIONHEART is detectable in human hearts and that Lionheart transcripts seem to have beneficial roles in cardiac

systolic function (Fig. 6). The molecular function of human LIONHEART might be different from mouse Lionheart because Myh6 mRNA has been detected at about 30% of the total Myh transcripts and MYH6 protein is ~10% of the total MYH protein in the adult human heart. The dominant type of MYH in human hearts is MYH7[5]. Conversely, MYH6 was detected at more than 90% in young adult rodents. Considering these, deciphering the molecular function of human LIONHEART might be required to examine whether gain-of-function of LIONHEART is beneficial in human heart failure with reduced ejection fraction.

In this report, we identified a lincRNA that is involved in cardiac remodeling induced by pressure overload. To the best of

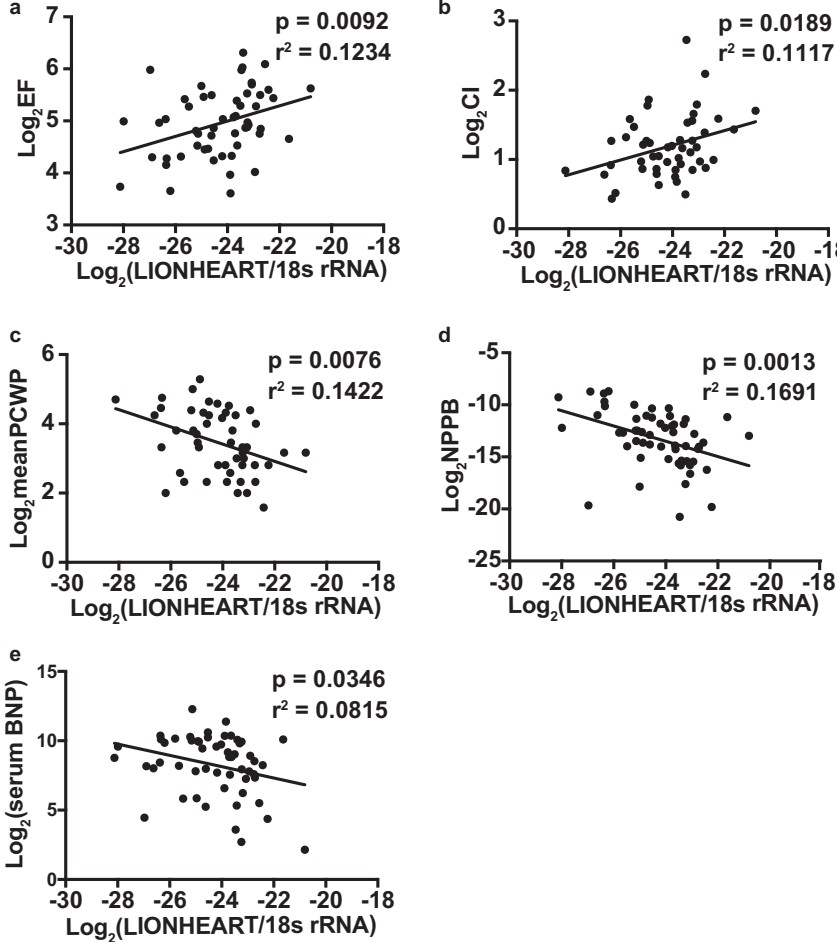

**Fig. 6 LIONHEART levels in human biopsy specimens and association study between the LIONHEART levels and clinical parameters. a** Association between human LIONHEART levels in cardiac left ventricular (LV) specimens and LV ejection fraction (EF). $n = 54$; $p = 0.0092$; $r^2 = 0.1234$. **b** Association between human LIONHEART levels and cardiac index (CI). $n = 49$; $p = 0.0189$; $r^2 = 0.1117$. **c** Association between human LIONHEART levels and mean pulmonary capillary wedge pressure (PCWP). $n = 49$; $p = 0.0076$; $r^2 = 0.1422$. **d** Association between human LIONHEART levels and NPPB levels in LV specimens. $n = 58$; $p = 0.0013$; $r^2 = 0.1691$. **e** Association between human LIONHEART levels and serum NPPB (BNP) levels. $n = 55$; $p = 0.0346$; $r^2 = 0.08148$.

our knowledge, Lionheart is the first example of a lincRNA that regulates *Myh6* expression. Given that noncoding DNA in the genome has expanded during evolution and these regions yield a lot of lincRNAs, these transcripts seem to constitute complex machineries in heart physiology and the pathophysiology of cardiovascular disease in humans. Uncovering the in vivo function of each lincRNA precisely may lead to novel potential therapies for human cardiovascular diseases.

## Methods

**Mice.** C57BL/6J mice were purchased from Charles River Laboratory. Pregnant C57BL/6J mice and neonatal C57BL/6J mice were also purchased from Charles River Laboratory. Specific-pathogen-free animals were maintained in the animal laboratories of Kyoto University Graduate School of Medicine, and the Kyoto University Ethics Review Board approved this investigation. Twelve-week-old male mice were treated with TAC, as described previously[25,26] for the screening of lincRNA deregulated during cardiac remodeling. Any mice that experienced complications from surgery were excluded. Phenylephrine and isoproterenol-induced cardiac hypertrophy was described previously[26]. All analyses were performed in a uniform and unbiased fashion. The generation of *Lionheart*-KO mice is described in Supplementary Methods.

**Generation of *Lionheart* knockout mice.** *Lionheart* knockout mice were generated based on the previous report[27] in which C57BL/6J mouse embryonic stem (ES) cells and homologous recombination system were used. The targeting vector was constructed by modifying bacterial artificial chromosome RP using defective

prophage λ-Red recombination system. As a selection marker, neomycin resistance cassette flanked by loxP sites (loxP-PGK-gb2-neo-loxP cassette: Gene Bridges) was inserted at the *Lionheart* locus. The targeting vector was electroporated into C57BL/6J mouse ES cells. Homologous recombination was confirmed by Southern blotting as shown in Supplementary Fig. 6b. Appropriately recombined ES cells were injected into blastcysts from BALB/c strain mice supplied by Unitech Inc, and the blastcytes were implanted into the uterus of ICR strain female mice to obtain the chimeric mice. The neomycin resistance cassette was removed in the mouse germ line by breeding the heterozygous mice with knocked-in *Ayu-1 Cre* recombinase expressed in ubiquitous tissues, including the germ line. Descendant *Lionheart* heterozygous mice without the *Ayu-1 Cre* allele were crossed with each other to generate *Lionheart*-deficient mice. All experiments were carried out in C57BL/6J background mice, and WT littermates were used as a control.

**Preparations of mouse organs.** Mice were euthanized at the indicated weeks after TAC or sham operation, and the hearts were excised. The hearts were washed immediately by cold phosphate-buffered saline (PBS), and weighted. Cut hearts were frozen with liquid nitrogen by snap frozen, and saved at −80 °C until RNA extraction. Normal organs were excised from 12-week-old male mice, and stored in the same manner.

**Primary neonatal mouse ventricular cardiomyocytes culture.** Primary neonatal mouse ventricular cardiomyocytes were isolated as previously described with some modification[28]. Briefly, hearts from ~40 neonatal C57BL/6J mice were excised, minced, and enzymatically digested in buffer containing pancreatin (Sigma), and collagenase type II (Life Technologies) for 30 min six times. Cells were washed by centrifugation in a 3:1 mixture of DMEM and Medium 199 (Life Technologies) supplemented with 10% horse serum and 10% fetal bovine serum, and collected.

Neonatal cardiomyocytes and non-cardiomyocytes were separated by differential plating for 40 min, and cardiomyocytes were mixed in 10 ml of culture medium. After the measurement of cell concentration, cardiomyocytes were expanded in culture medium (300,000/ml), and 5-bromo-2′-deoxyuridine (final concentration: 0.1 mM, Sigma-Aldrich) was added to prevent the proliferation of non-cardiomyocytes. Cells were plated in Primaria™ 6-well plates (Becton Dickinson), and cultured in a 5% $CO_2$ incubator at 37 °C. Twenty-four hours later, cardiomyocytes were washed by serum free medium twice, and stimulated with angiotensin II or phenylephrine at indicated concentrations.

**Separation of neonatal cardiac myocytes and fibroblasts**. Neonatal mouse cardiac myocytes and cardiac fibroblasts were separated using fluorescent-activated cell sorting (FACS) as previously described[29]. Neonatal rat cardiac myocytes and cardiac fibroblasts were separated as previously described[26,29].

**Human iPSCs and iPSCs-derived cardiac myocytes**. Human iPSCs and human iPSCs-derived cardiac myocytes were obtained as previously described[30]. At 21 days after induction of cardiac myocytes, iPSCs-derived cardiac myocytes were sorted with GFP using fluorescence-activated cell sorting. Total RNAs in those cells were isolated as described below.

**Left ventricular myocardial biopsy**. We increased the number of myocardial biopsy cohort that is examined in previous report[26,31], and the number reached 58 patients. All patients provided written informed consent for the procedure and gene expression analyses. The Ethics Committees of Osaka Red Cross Hospital and Kyoto University Hospital approved the study protocol. The ejection fraction was calculated by left ventriculography. The hemodynamic parameters were collected by Swan-Ganz catheterization. When there was missing data in echocardiography or hemodynamic parameters, the patient was omitted from the analysis in Fig. 6. Total RNA extraction from the specimens was performed using 1 mL of TRIzol® reagent (Invitrogen). Patient characteristics were listed in Supplementary Table 3.

**RNA extraction**. To extract total RNA, the organs were homogenized in 1 mL of TRIzol® reagent (Invitrogen) using homogenizer, and the total RNA was extracted in accordance with the manufacture's protocol. Total RNA extracted from cells was also isolated using 1 mL of TRIzol® reagent (Invitrogen). The quantity and quality of total RNA were determined using Nanodrop (Thermo Scientific).

**Microarray analysis**. To screening of deregulated lincRNAs, each 500 ng of total RNA extracted from 4 to 6 mice were mixed, and analyzed by microarray analysis (SurePrint G3 Mouse GE 8x60K, Agilent Technologies). Analysis of microarray data was carried out by the GeneSpring GX11 software (Agilent Technologies). The data were deposited in GEO repository and the GEO accession number is GSE153814.

To assess the global expression changes of previously identified lncRNAs in Lionheart-KO mouse hearts after TAC surgery, each 3 μg of total RNA extracted from six control mice and each 2 μg of total RNA from 12 Lionheart-KO mice were mixed and analyzed by microarray analysis (GeneChip Mouse Gene 2.0 ST Array, Affymetrix). The RNA quality was evaluated with Agilent 2100 Bioanalyzer (Agilent Technologies, Inc). For the microarray analysis, sense-strand DNA targets were generated using GeneChip WT PLUS Reagent Kit (Thermo Fisher Scientific). Briefly, reverse transcription reaction was performed using a primer containing T7 promoter and thermal cycler (Veriti 200, Life Technologies), and cDNA was synthesized from 100 ng of total RNA. Antisense cRNA was synthesized from the obtained cDNA by in vitro transcription reaction with T7 RNA polymerase. After amplification and purification, sense cDNA was synthesized from 15 μg of cRNA using random primers and the cDNA was purified with RNase H. Using Uracil-DNA glycosylase, 5.5 μg of cDNA was fragmented and then biotinylated. Biotinylated cDNA was injected into GeneChip Mouse Gene 2.0 ST Array (Affymetrix) and hybridized with probe using a GeneChip hybridization instrument (Hybridization Oven 645, Affymetrix). After hybridization, the array was washed and stained with phycoerythrin by GeneChip Hybridization, Wash, and Stain Kit and Fluidics Station 450 (Affymetrix). The fluorescent signal was scanned with Scanner 3000 7G (Affymetrix) and the result was obtained by GeneChip Command Console. Analysis of microarray data was carried out by Affymetrix Transcript Analysis Console software (Affymetrix) and scatter plots were created using R ver. 3.5.3 (R Foundation for Statistical Computing).

**Quantitative reverse transcription-PCR (qRT-PCR)**. Total RNA was reverse transcribed by oligo-dT primers or blend primer of random hexamers and oligo-dT 3:1 (volume/volume) in accordance with the manufacture's protocol (Transcriptor First Strand cDNA Synthesis Kit, Roche), and the cDNA was analyzed by qRT-PCR. Amplification by FastStart Universal SYBR Green Master (Roche) or THUNDERBIRD® SYBR qPCR Mix (TOYOBO Life Science) of each sample was duplicated, and the qReal-Time PCR was run for 40 cycles using 7900HT Fast Real-Time PCR System (Applied Biosystems). To design the upregulated lincRNA specific primers, we affirmed the regions of each lincRNAs in chromosomes and probe sequences (60 nucleotides) of microarray. Several hundred nucleotides

containing the probe sequences were inserted the Primer3 (http://primer3.wi.mit.edu), which is a widely used program for designing PCR primers, and we designed the primers for each lincRNAs. Specificity was confirmed by the dissociation curves in qRT-PCR results. Expressions of lincRNAs and messenger RNAs (mRNAs) were normalized by the housekeeping gene *Actb* or *18s ribosomal RNA* (*18s rRNA*), and calculated by the $2^{-\Delta\Delta Ct}$ method. Primer sequences used in qRT-PCR are follows. mmu-Lionheart: 5′-GAGGCGAGAAGTGCTTGTAGGA-3′ (Forward), 5′-AAGAACTTCTGCTCGGAGGACC-3′ (Reverse); rno-Lionheart: 5′-CTGGGAGAGGCAAGAAGTGTTT-3′ (Forward), 5′-GAGGAGCCAGTTGAACTCAGAG-3′ (Reverse); hsa-LIONHEART: 5′-AAGAGGTGAGAAGCTGCTTGAA-3′ (Forward), 5′-CCAGTTGAATACCGAGAATGGT-3′ (Reverse); mmu-Myh6: 5′-TCCGAAAGTCAGAGAAGGAACG-3′ (Forward), 5′-ACACGACCTTGGCCT-TAACATA-3′ (Reverse); mmu-Myh7: 5′-ATTCTCCTGCTGTTTCCTTACTTG-3′ (Forward), 5′-TTGGATTCTCAAACGTGTCTAGTG-3′ (Reverse); mmu-Sestd1: 5′-TGTGTCATTCTCCCATCAGCGT-3′ (Forward), 5′-ACACTGAAGTCATCTCCACGGG-3′ (Reverse); mmu-Zfp385b: 5′-GGAGGCTCACAACACAGGATCT-3′ (Forward), 5′-GTAGTCCTGAGCCCTTACTGCC-3′ (Reverse); mmu-Nppb: 5′-GCCAGTCTCCAGAGCAATTCA-3′ (Forward), 5′-TGTTCTTTTGTGAGGCCTTGG-3′ (Reverse); mmu-Col1a1: 5′-GCCAAGAAGACATCCCTGAAG-3′ (Forward), 5′-TCATTGCATTGCACGTCATC-3′ (Reverse); mmu-Actb: 5′-AGATTACTGCTCTGGCTCCTA-3′ (Forward), 5′-CAAAGAAAGGGTGTAAAACG-3′ (Reverse). mmu-Acta2: 5′-CACCGCAAATGCTTCTAAGT-3′ (Forward), 5′-GGCAGGAATGATTTGGAAAGG-3′ (Reverse); hsa-ACTB: 5′-AGGCACTCTTCCAGCCTTCC-3′ (Forward), 5′-GCACTGTGTTGGCGTACAGG-3′ (Reverse); mmu-*Mhrt*: 5′-GAGCATTTGGGGATGGTATAC-3′ (Forward), 5′-TCTGCTTCATTGCCTCTGTTT-3′ (Reverse); mmu-Chast: 5′-CCACTGACCCT-CATCCTTGT-3′ (Forward), 5′-CCCAGAAAGTGCCTCCTTTGT-3′ (Reverse); mmu-Chaer: 5′-TCCAATGAGGGAAGCGAAGC-3′ (Forward), 5′-GTCCGATGCCAGTTCCAGTT-3′ (Reverse); mmu-CHRF: 5′-CAACTTTACCCATCTCTTCTC-3′ (Forward), 5′-CTGAATTACTTCAGAGGAAAG-3′ (Reverse).

**Cloning of Lionheart**. To identify the transcription starts site (TSS) and the polyA sites of Lionheart, we carried out the 5′RNA ligase mediated rapid amplification of cDNA ends (5′RLM-RACE) and 3′RACE using the First Choice® RLM-RACE kit (Ambion) in accordance with manufacturer's instructions. PCR products were cloned into pCR2.1-TOPO using TOPO® TA® cloning kit (Invitrogen), and sequenced. The longest Lionheart transcripts was cloned. Primer sequences for cloning are 5′-AAAGTAGGACAAGTAACTGAAGC-3′ (Forward) and 5′-GAGTTTTAGATTTTTATTTAAGGG-3′ (Reverse).

**In vitro translation assay**. TNT® Quick Coupled Transcription/Translation System (Promega) and Transcend™ Non-Radioactive Translation Detection Systems (Promega) were used in accordance with the manufacture's protocol. The product was resolved directly on an SDS-PAGE gel, transferred to a nitrocellulose membrane, and detected by chemiluminescent reaction.

**Isolation of cytoplasmic and nuclear RNAs from mouse heart**. To extract the cytoplasmic and nuclear RNA from adult mouse heart, we used Cytoplasmic and Nuclear RNA Purification Kit (NORGEN Biotek Corporation)[14]. The normal hearts were excised from C57BL/6J mice at 10 weeks of age.

**Plasmids**. For promoter assay, we used pGL3-Enhancer vector (Promega). Indicated promoters were cloned from genome obtained from normal mouse heart, and the sequences were inserted into the upstream of luciferase in pGL3-Enhancer vector. For mutagenesis of the promoters, we used QuikChange Site-Directed Mutagenesis kit (Agilent Technologies) in accordance with the manufacture's protocol. pRL-TK™ Renilla reniformis luciferase plasmid (Promega) was used as the transfection efficiency control[32].

**Lentivirus production and DNA transduction**. Lentiviral stocks were produced in 293T cells in accordance with the manufacturer's instructions (Invitrogen), as described previously. In brief, virus-containing medium was collected for 48 h after transfection and filtered through a 0.45-μm filter. One round of lentiviral infection was performed by replacing the medium with virus-containing medium that contained 8 μg/mL polybrene, followed by centrifugation at $1220 \times g$ for 30 min.

**Dual-luciferase assay**. Reporter vectors were transfected into 293T cells or NMCMs using *Trans*IT®-LT1 reagent (Mirus Bio LLC). After 2 days incubation, both luciferase activities were measured using a dual-luciferase reporter assay system (Toyo Ink) as previously described[29,32].

**Echocardiography**. We used Vevo® 2100 (VISUALSONICS) at the indicated time points for assessment of cardiac function[29]. Mice were kept under inhalation anesthesia with isoflurane, and the heart rates were kept at 480–500 beats/min.

**Measurement of blood pressure**. Control mice and Lionheart-KO mice at 8 weeks after TAC surgery were used for blood pressure (BP) measurement. The

mouse was stayed in a mouse-holder (BP98-NTMm, Softron) without anesthesia and prewarmed at 37 °C (THC-31, Softron) for at least 2 min. The systolic, mean, and diastolic BP were measured by a programmable sphygmomanometer (BP-98A, Softron) using the indirect tail-cuff method. Blood pressure was measured 7–11 times in each mouse and the averages were used for the analysis.

**WGA staining and the quantification**. Paraffin-embedded ventricular short-axis sections were stained with FITC-conjugated lectin (Sigma, L4895). Images were captured using BZ-9000 (Keyence). Cardiomyocyte cross-sectional area measurement was performed using ImageJ64 software (NIH). Approximately 150–200 cells were measured per heart at ×400 magnification, and averages were used for the analysis.

**Western blotting analysis**. Western blotting was performed as previously described[29]. A total of 30 μg protein was fractionated using NuPAGE™ 4–12% Bis-Tris (Invitrogen) gels and transferred to a Protran™ nitrocellulose transfer membrane (Whatman). The following primary antibodies were used: anti-glyceraldehyde-3-phosphate dehydrogenase (GAPDH) (Cell Signaling, 14C10), 1:3000; anti-heavy chain cardiac myosin (Abcam, BA-G5, ab50967) for MYH6, 1:1000; monoclonal anti-Myosin (Skeletal, Slow) for MYH7 (Sigma-Aldrich, M8421), 1:1000. For captured proteins by RNA pulldown or RNA immunoprecipitation, anti-PURA (Santa Cruz, 80-L, sc-130397) and anti-CSRP3 (Abcam, ab173301) were used. Anti-rabbit IgG (GE Healthcare) or anti-mouse IgG (GE Healthcare) were used as secondary antibodies each at a dilution of 1:2000. After final wash in 0.05% T-PBS (1× PBS and 0.05% Tween-20), the immune complexes were detected using the Pierce™ ECL or ECL Plus Western Blotting Substrate. Immunoblots were detected using LAS-3000 (Fujifilm). For quantification of western blotting, densitometric analyses were performed using ImageJ64 software (NIH). All uncropped images are shown in Supplementary Figs. 14 and 15.

**Transmission electron microscopy**. Left ventricular hearts were cut into 1-mm cubes after perfusion fixation and were immediately post-fixed in a mixture of 4% paraformaldehyde and 2% glutaraldehyde overnight at 4 °C. The specimens were further fixed by immersing in 1% osmium tetroxide. The specimens were then dehydrated with a graded series of ethanol and embedded in epoxy resin. Ultrathin sections were cut with an ultramicrotome and double-stained with uranyl acetate and lead citrate and then examined by transmission electron microscopy (H-7650, Hitachi). Cardiomyocytes containing the correct orientation of Z-lines were imaged for mitochondria. Mitochondria size was measured using ImageJ64 (NIH). Average sizes were used for the analysis.

**Preparation of biotin-labeled RNAs**. To obtain the biotin-labeled full-length or truncated Lionheart, we used CUGA® 7 in vitro Transcription kit (NIPPON GENE). When the biotin-labeled RNAs were synthesized, biotin-16-UTP (Sigma-Aldrich) was added, as the ratio of biotin-16-UTP to usual UTP was 3:2. The PCR product containing T7 promoter was employed as the template. The quantity and the quality of the synthesized RNAs were confirmed by NanoDrop™ 2000 spectrophotometer (Thermo Fisher Scientific) and Agilent 2100 bioanalyzer (Agilent Technologies). The RNAs were saved at −80 °C until use. To identify the Lionheart-specific-binding proteins, we used the biotin-labeled Lionheart antisense (AS) as the control.

**RNA pull-down assay**. Approximately 60 mg of TAC heart was used for each biotin-labeled RNAs. The stocked 10× hypotonic lysis buffer (100 mM HEPES pH 7.9, 15 mM MgCl$_2$, 100 mM KCl) was diluted in RNase DNase free water and supplemented with the protease inhibitor cocktail (cOmplete™ Mini Protease Inhibitor Cocktail, Roche), RNase inhibitor (TOYOBO), phenylmethylsulfonyl fluoride (PMSF, Roche, final concentration: 1 mM), and Dithiothreitol (DTT, final concentration: 1 mM) just before use. The hypertrophied heart was homogenized in 1 ml of the 1× hypotonic lysis buffer using a dounce homogenizer, and the suspension was centrifuged by 10,500 × g for 20 min at 4 °C. The supernatant was transferred to the new tube and frozen in liquid nitrogen as the cytoplasmic extract. The pellet was resuspended in 1 ml RIP buffer (150 mM KCl, 25 mM Tris-HCl pH 7.4, 0.5% NP40). The protease inhibitor cocktail, RNase inhibitor, PMSF (final concentration: 1 mM), and DTT (final concentration: 0.5 mM) were supplemented just before use. The resuspended pellet was sonicated by Bioruptor® (Cosmo Bio) in high setting for 45 min (30-s on, 30-s off), and centrifuged at 15,300 × g for 10 min. The supernatant was transferred to the new tube, and frozen in liquid nitrogen as the nuclear extract. The supernatant was saved at −80 °C until use. Dynabeads® M-280 streptavidin (Thermo Fisher Scientific) was prepared for RNA pull-down assay in accordance with the manufacture's protocol, and the beads were washed three times by RIP-IB buffer (20 mM Tris pH 7.5, 1 mM EDTA, 150 mM NaCl, 0.1% NP40, 5% Glycerol). The protease inhibitor cocktail, RNase inhibitor, and DTT (final concentration: 0.5 mM) were supplemented to RIP-IB buffer just before use.

To allow for proper folding of the biotin-labeled RNAs, 100 pmol of RNA was added into 100 μl of RNA structure IB buffer (10 mM Tris pH 7.0, 0.1 M KCl, 10 mM MgCl$_2$), heated to 90 °C, and cooled down to 4 °C. The RNA was mixed with streptavidin magnetic beads in 300 μl of RIP-IB buffer supplemented with protease

inhibitor cocktail, RNase inhibitor, and DTT. Then, the mixture of RNA and beads was rotated for 30 min at RT, and for 30 min at 4 °C.

To reduce the nonspecific binding of nuclear proteins to beads, the nuclear extract was precleared by incubation with washed magnetic beads for 1 h at 4 °C with rotation. The beads binding with RNA were washed by 400 μl of RIP-IB buffer supplemented with protease inhibitor cocktail, RNase inhibitor, and DTT. Then, the precleaned extract was mixed with the beads, and incubated with rotation for at least 4 h at 4 °C. After the incubation, the beads were washed by 600 μl of RIP buffer supplemented with protease inhibitor cocktail, RNase inhibitor, and DTT. The captured protein was obtained from the beads by boiling in SDS buffer.

**Mass spectrometry**. To identify the unbiased Lionheart-binding protein, all Lionheart-binding proteins were identified by mass spectrometry. Technically, the Lionheart-binding nuclear or cytoplasmic proteins were shortly electrophoresed into SDS-polyacrylamide gel, stained with Coomassie Brilliant Blue (CBB Stain One, Nacalai Tesque), and the area from the well-bottom to the dye-front was excised out for the collection of the proteins. Similarly, the biotin-labeled Lionheart AS-binding nuclear or cytosolic proteins were collected as the controls. The proteins in-gel pieces were digested and recovered using In-gel Tryptic Digestion Kit for Mass Spectrometry (Thermo Fisher Scientific) according with the manufacturer's instruction. The recovered peptides were resuspended in 0.1% formic acid and separated using nano-flow liquid chromatography (Nano-LC-Ultra 2Dplus System, Eksigent, Dublin, CA), which was used in a trap and elute mode with trap column (200 μm × 0.5 mm ChromXP C18-CL 3 μm 120 Å, Eksigent) and analytical column (75 μm × 15 cm ChromXP C18-CL 3 μm 120 Å, Eksigent). The separation was carried out using a binary gradient with solvent A (0.1% formic acid) and B (0.1% formic acid, 80% acetonitrile). The gradient program used was as follows: 2–40% B in 125 min, 40–90% B in 1 min, 90% B for 5 min, 90–2% B in 0.1 min, 2% B for 18.9 min, at 300 nL/min. The eluates from nano-LC were directly infused to the mass spectrometer (TripleTOF 5600+ system, SCIEX, Framingham, MA). The datasets were acquired with the information-dependent acquisition method. The identification of peptides/proteins was carried out using ProteinPilot software version 4.5beta (SCIEX) with UniProtKB/Swiss-Prot database (Mus musculus, June 2014) appended with known contaminant database (SCIEX). The relative abundances of the identified proteins were estimated through label-free quantification using Progenesis QI for Poteomics software (Nonlinear Dynamics). The relative abundance of each protein was calculated by the grouping of non-conflicting peptides and the results for proteins identified by at least two distinct peptides having at least 95% confidence were used. The proteins which were enriched in the Lionheart-S-binding proteins compared with the AS-binding proteins (more than 1.5-fold) were considered as the Lionheart-specific-binding proteins, and analyzed by DAVID Bioinformatics Resources 6.8 (https://david.ncifcrf.gov/home.jsp).

**RNA immunoprecipitation (RIP)**. RIP assay was based on the previous report[16]. One hundred forty microliters of formaldehyde (FA) was added to 4.85 ml of PBS, and the FA solution was transferred to a glass dish on ice. For cross-linking, heart tissue was minced in the FA solution by scissors and incubated at room temperature for 15 min with rotation. The minced tissue was transferred to a new tube, and the tube was centrifuged by 1250 × g for 3 min at 4 °C. We discard the supernatant and added 5 ml of 0.125 mol/L of glycine solution. After centrifugation by 1250 × g for 3 min at 4 °C, the tissue was washed twice by cold PBS supplemented by PMSF. Finally, the cross-linked heart tissue was centrifuged by 800 × g for 10 min at 4 °C. The supernatant was discarded and the cross-linked heart tissue was saved at −80 °C until use.

The stocked 10× hypotonic lysis buffer was diluted in RNase DNase free water, and protease inhibitor cocktail, RNase inhibitor (TOYOBO), DTT (final concentration: 1 mM), and NP40 (final concentration: 0.5%) were supplemented just before use. The heart tissue was homogenized in the 1× hypotonic lysis buffer using a dounce homogenizer, and the solution was centrifuged by 10,500 × g for 20 min at 4 °C. The supernatant was transferred to the new tube and frozen in liquid nitrogen as the cytoplasmic extract. The nuclear protein complex was obtained by the same manner as RNA pull-down assay described above. The nuclear extract was aliquoted and the antibody for the RNA-binding protein or control Ig was added. The mixtures were incubated for more than 6 h at 4 °C with rotation. Small amount of nuclear extract was saved as the input sample.

Based on the manufacture's protocol, Dynabeads protein G (Thermo Fisher Scientific) was washed and the mixture of antibody and the nuclear extract was added to the beads. The mixture of beads, antibody, and the nuclear extract was incubated for 1 h at 4 °C with rotation. After incubation, the supernatant was removed, and the beads was washed by RIP buffer three times supplemented with the protease inhibitor cocktail, RNase inhibitor, PMSF, and DTT just before use. The beads were divided for subsequent applications.

Before RNA extraction, the beads were treated with proteinase K for 1 h at 65 °C followed by the treatment of DNase for 15 min at room temperature. To extract the RNA, we used 1 mL of TRIzol® reagent (Invitrogen). Using the isolated RNA, we generated cDNA as described above, and determined the contained transcripts levels by qRT-PCR. When protein extraction, the beads were treated with DNase, RNase H (New England BioLabs), and RNase A (Sigma-Aldrich) for 30 min at

37 °C. The captured protein was obtained from the beads by boiling in SDS buffer, and analyzed by western blotting.

**Chromatin immunoprecipitation (ChIP).** Cross-linked nuclear protein–DNA complexes from TAC or sham operated mouse heart were extracted as described in the methods of RIP section. For ChIP, anti-PURA antibody (Santa Cruz, 80-L, sc-130397) was used. The nuclear complexes were mixed with the antibodies with rotation at 4 °C for at least 6 h. Tenth part of the complexes was saved as the input at −80 °C until use. The mixture of antibody and the cross-linked nuclear extract was added to the washed Dynabeads protein G and incubated for 1 h at 4 °C. The beads were washed four times by RIP buffer in which the protease inhibitor cocktail, PMSF, and DTT were added just before use. After that, the beads were treated with RNase H, and RNase A for 30 min at 37 °C twice followed by the treatment with Proteinase K (Sigma-Aldrich) for 1 h at 65 °C. To elute the DNA, the solution containing the beads treated with RNase H, RNase A, and Proteinase K was treated with TE saturated phenol, incubated at room temperature for 30 min with rotation, and centrifuged at $10,000 \times g$ for 5 min. Upper layer was transferred to new tube, and the DNA was precipitated by isopropanol. Quantification of PNR element in *Myh6* first intron was performed using 7900HT Fast Real-Time PCR System (Applied Biosystems). Primer sequences are 5′-CCAACCCAGGTAAG AGGGAGTTTC-3′ (Forward) and 5′-AACTTCCCAGGCTGGTGGAAGG-3′ (Reverse).

**RNA electrophoretic mobility shift assay (EMSA).** Single-strand PNR probe with 5′ biotin-label was designed and purchased (Thermo Fisher Scientific). To obtain purified PURA protein, PURA was cloned into a CMV-driven expression vector with His-tag. After transfection into 293T cells using standard PEI (Poly-sciences)-mediated transfection method, extracted proteins were purified using MagZ protein purification system (Promega). To obtain full-length Lionheart, we used RiboMAX Large Scale RNA Production Systems (Promega). RNA EMSA was performed by using the LightShift Chemiluminescent RNA EMSA Kit (Thermo Fisher Scientific) according to the manufacturer's instructions. The analytical binding reaction was carried out in a total volume of 20 μl containing 10 fmol of the labeled DNA probe, 150 pmol of PURA protein, 0.05% NP40, and 1 μg of poly (dI-dC) as nonspecific competitor. Lionheart RNA or unlabeled probe at indicated concentrations were used for competition experiments. After incubation at room temperature for 20 min, the reaction mixtures were loaded onto 6% DNA Retar-dation Gels (Thermo Fisher Scientific), transferred to Biodyne B Nylon Membranes (Thermo Fisher Scientific). The membrane was then cross-linked by exposure to UV (120 mJ/cm²). After blocking the membrane, the biotin-labeled probes were detected using Amersham Imager 680 (GE Healthcare Life Sciences). For quanti-fication of the acquired images, densitometric analyses were performed using ImageJ64 software (NIH) and GraphPad Prism 7 statistical packages. The software facilitates the fitting nonlinear regression model.

**Generation of AAV9 vector and the injection.** AAV vector was generated using AAV-2 Helper-Free System in accordance with the manufacturer's protocol (Cell Biolabs). Plasmid of AAV capsid serotype 9 was obtained from Penn Vector Core at University of Pennsylvania. We used pAAV-MCS as control. AAV-293 cells (Agilent Technologies) were transfected with pAAV-MCS or pAAV-*Lionheart*, pHelper, and pAAV-RC9 vector plasmids using standard PEI (Polysciences)-mediated transfection method. Seventy-two hours after transfection, cells were collected and suspended in buffer solution containing 124 mM NaCl, 3 mM KCl, 26 mM NaHCO₃, 2 mM CaCl₂, 1 mM MgSO₄, 1.25 mM KH₂PO₄, and 10 mM D-Glucose. Following four freeze-thaw cycles, the cell lysates were treated with Benzonase nuclease (Milliore) at 45 °C for 15 min, then centrifuged twice at $16,000 \times g$ for 10 min at 4 °C. The supernatant was used as the virus-containing solution. Quantitative real-time PCR was performed to measure the titer of the purified virus[33]. Virus aliquots were then stored at −80 °C until used for the experiment.

At postnatal day 3 (P3), *Lionheart*-KO mice were injected with AAV9-Control or AAV9-*Lionheart* at a dose of $5 \times 10^{10}$ virus genomes per animal by intraperitoneal injection. At the age of 8 weeks old, male mice underwent TAC surgeries. Animals were analyzed at 8 weeks after TAC operation.

**Bioinformatic analysis.** To predict secondary structure of Lionheart, RNAfold (http://rna.tbi.univie.ac.at/cgi-bin/RNAWebSuite/RNAfold.cgi) was used. ChIP-seq data using SRF antibody was obtained from Active Motif website (http://www.activemotif.com/catalog/details/61385/srf-antibody-mab-clone-2c5). GO analysis for identified Lionheart-specific binding proteins was performed by DAVID bioinformatic resources 6.8 (https://david.ncifcrf.gov/home.jsp). Accessibilities in Lionheart were computed using ParasoR[34] with the Turner energy model of Vienna RNA Package 2.0. Local purine densities of Lionheart were computed for all sequence windows of 21 nucleotides. Then, they were converted to quantile values so that they fit in the range of 0–1. Interaction between Lionheart and PURA was analyzed by *cat*RAPID (http://s.tartaglialab.com/page/catrapid_group).

**Statistics and reproducibility.** In vivo experiments, measurements were taken from distinct samples and sample numbers indicate biologically independent animal numbers. Sample sizes were determined by power calculations based on effect sizes previously reported in the literature. In all studies including in vivo and in vitro studies, data were obtained from at least three biological replicates. For quantification of western blotting, densitometric analyses were performed using ImageJ64 software (NIH). All uncropped blot images are shown in Supplementary Figs. 14, 15. Error bars are defined as SEM (standard error of the mean). For statistical comparisons, Mann–Whitney test (2 groups) and one-way ANOVA (*n* groups) with Bonferroni's post hoc test were used as appropriate. For calculations of *p*-value with Mann–Whitney test, two-tailed test was used. To compare echo-cardiography assessment observed overtime, two-way ANOVA with Bonferroni's post hoc test was used. For association studies with human heart biopsy specimens, linear regression was used. The *p*-value < 0.05 was considered to indicate statistical significance. Statistical analyses were performed using GraphPad Prism 7 statistical packages. $*p < 0.05$; $**p < 0.01$; $***p < 0.001$; $###p < 0.001$.

**Reporting summary.** Further information on research design is available in the Nature Research Reporting Summary linked to this article.

## Data availability

All data of this study are shown in the main text and supplementary information files. The source data underlying the figures are presented in Supplementary data file except Fig. 1a data. The Fig. 1a data is deposited in GEO repository and the GEO accession number is GSE153814. Any additional source data or materials used in this study can be obtained from the corresponding author upon reasonable request.

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

## Acknowledgements
We are grateful to Justin Boyer for critical reading of the paper. This work was supported by the Ministry of Education, Culture, Sports, Science and Technology (MEXT) and Japan Society for the Promotion of Science (JSPS) KAKENHI Grant that numbers are JP16K19400 (Y.K.), JP18J20214 (S.T.), JP1605297 (T.K.), and JP17H05599 (K.O.). This study was also supported by SENSHIN Medical Research Foundation, Japan Heart Foundation Research Grant, Takeda Science Foundation, and Kowa Life Science Foundation for Y.K., and by the Uehara Memorial Foundation for M.N.

## Author contributions
K.O. managed this project. Y.K. and S.T. designed and performed most in vitro and in vivo experiments. M.N., S.I., S.W., and S.K. performed lincRNA screening and RNA pull-down screening. K.N. and T.I. conducted human left ventricular biopsies and made the database. Takahiro Horie supported the generation of *Lionheart*-KO mice. Y.K. and H.K. conducted bioinformatic analyses. Takeshi Hatani and Y.Y. carried out the experiments using human iPSCs. M.I., Y.N., and N.S. performed NMCM isolation and RNA extraction from embryonic mouse hearts. O.B., Tetsushi Nakao, Tomohiro Nishino, Y.I., and F.N. provided intellectual input on the project. Y.M. and M.K. helped the experiments including promoter assay and AAV9 production. T.K. supported the project financially. Y.K., S.T., and K.O. wrote the paper.

## Competing interests
The authors declare no competing interests.
