## [Peer Review File · Communications Biology]

Reviewers' comments:

Reviewer #1 (Remarks to the Author):

In this manuscript, the authors report the results of experiments demonstrating that there are multiple long intergenic non-coding RNAs (lincRNAs) that are upregulated in the (mouse) left ventricles with pressure overload cardiac hypertrophy, induced by banding of the transverse thoracic aorta. Specifically, they report the identification of a novel, striated muscle-abundant lincRNA which they have termed Lionheart. Following the identification of the upregulation of Lionheart with banding, the authors generated a mouse line in which Lionheart was eliminated (deleted). They show that pressure overload in these targeted deletion animals resulted in decreased systolic function, compared with control animals. In addition, they demonstrate that Lionheart interacts with the Purine-rich element-binding protein A (PURA) and prevents PURA from binding to the promoter region of the Myh6 locus, thereby resulting in reduced Myh6 transcript and MYH6 protein expression. Finally, the authors present data showing that human LIONHEART levels in left ventricular biopsy specimens are positively correlated with cardiac systolic function. The data presented appear to be of high quality and there are no major technical concerns regarding data acquisition, analysis or presentation. The authors conclude then that the LIONHEART lincRNA plays an important role in cardiac remodeling with pressure overload and that this results from the regulation of MYH6 expression. While this is a reasonable conclusion based on the experiments completed and the data presented, it is unclear from the data presented how many other RNAs (of various types) are affected and/or how the dysregulated RNAs identified here compare to those identified in previous studies. In addition, the authors have not provided any direct information about, or insights into, the molecular/cellular/systemic mechanisms involved in mediating the functional effects of Lionheart. For example, it is not clear how pressure overload affects PURA expression and, in turn, how this affects Lionheart/LIONHEART expression? It is unclear if the observed changes are cell/region specific and/or if there are systemic changes in PURA and Lionheart/LIONHEART. Are there changes in these proteins/transcripts in the right ventricles following pressure overload induced cardiac hypertrophy? Also, it is unclear how Lionheart is linked to the various effects of pressure overload, including cellular hypertrophy and electrical remodeling, and tissue fibrosis, as these questions were not directly addressed in the studies reported in this manuscript. Much more could be done to explore/define the underlying mechanisms to increase the information/insights provided and to increase the impact of the work.

Reviewer #2 (Remarks to the Author):

Kuwabara et al. found upregulated lincRNAs obtained from TAC-treated mouse hearts as compared with control hearts and identified long intergenic-origin noncoding RNA in heart (lionheart). It is increased not only in mouse TAC model, but also angiotensin II- or phenylephrine-induced hypertrophied cardiomyocytes, is annotated at mm10_chr.2:77314725-77317068 on the minus strand, and is regulated transcriptionally by SRF via the promoter of the lionheart gene. The authors generated knockout (KO) mice, and observed structural and functional differences treated with TAC after 8 weeks in KO hearts, where Myh6, but not Myh7, was transcriptionally and translationally reduced. The authors tried to identify lionheart-binding proteins in mouse nuclear extracts of TAC-treated hearts and of 46 candidates, due to the highest specificity to lionheart, they focused on Purine-rich element-binding protein A (PURA), a negative regulator of Myh6 transcription by binding to a purine-rich negative regulatory (PNR) element in the first intron of the Myh6 gene. They also conducted the rescue experiments using AAV9-lionheart with lionheart KO mice treated by TAC. Finally, the authors measured the lionheart levels in human biopsy specimens and showed positive and negative correlations with reduced cardiac systolic functions.

Major concerns:

Phenylephrine, isoproterenol, and angiotensin II are regulators acting not only to cardiomyocytes, but also to blood pressure. How is the blood pressure, as a basic information of physiology of lionheart KO mice?

It is of interest that lionheart binds to PURA with highest specificity in hearts. As the authors mention, lionheart may a regulator of PURA to reduce the binding affinity to PNR. Thus, in lionheart KO mouse heart, PURA is able to bind to PNR with higher affinity than control heart, resulting in the repression of target genes, for example, Myh6. Is lionheart only one linc RNA that binds to PURA? If not, is there any RNA binding to PURA in lionheart KO mice, because of the lack of lionheart that occupies PURA with the highest affinity?

Although the authors hypothesize that lionheart binds to PURA, it may be increased in lionheart KO hearts as a negative feedback, because it lacks the binding partner with the high affinity. Did the authors check the PURA level, for example, by Western blotting?

It is well known that mitochondria play important roles in cardiac functions. The authors mention in p. 7, l. 3~12, lionheart may be involved in mitochondria. Are there any structural or biochemical changes in mitochondria of KO mice with/without TAC?

Minor concerns:

In p. 10, l. 7, driven via may be better than driven by because of the redundant usage of by.

Response to Reviewer #1

We are grateful to Reviewer #1 for the informative and useful comments. As described below, we have considered all of these comments and used them to improve our manuscript.

-Major

1. The authors conclude then that the LIONHEART lincRNA plays an important role in cardiac remodeling with pressure overload and that this results from the regulation of MYH6 expression. While this is a reasonable conclusion based on the experiments completed and the data presented, it is unclear from the data presented how many other RNAs (of various types) are affected and/or how the dysregulated RNAs identified here compare to those identified in previous studies.

We thank the reviewer for highlighting this important point. To assess the expression changes of other previously identified RNAs in *Lionheart*-KO mouse hearts after TAC surgery, we performed microarray analysis for the levels of primary-miRNAs (pri-miRNAs) and other long noncoding RNAs (lncRNAs). The levels of pri-miR-669a-3, pri-miR-709, pri-miR-1946b, and pri-miR-3962 were more than 2-times higher in *Lionheart*-KO mice than control mice, as shown in Supplementary Fig. 10a. No pri-miRNAs were down-regulated (fold change < 0.5) in *Lionheart*-KO mice compared with control mice. Among the upregulated miRNAs, it is reported that long-term miR-669a expression with AAV9 ameliorates cardiac systolic dysfunction in a dystrophic mouse model (Quattrocelli M., et al. J Am Heart Assoc. 2013;2:e000284). Thus, we are speculating that the miR-669a upregulation in *Lionheart*-KO mice is a compensatory machinery against cardiac systolic dysfunction observed in *Lionheart*-KO mice with pressure overload. Concerning the other lncRNAs, 5 lncRNAs were increased (fold change > 2) and 2 lncRNAs were decreased (fold change < 0.5) in *Lionheart*-KO mice compared with control mice, as shown in Supplementary Fig. 10b. Among these deregulated 7 lncRNAs, a report showed that *Snora75*, a noncoding small nucleolar RNA, is negatively regulated by miR-24 which is enriched in platelet-derived microparticles (PMPs). While the PMPs treatment led to tumor cell apoptosis resulting in inhibition of cancer cell growth in *vivo* and in *vitro* (Michael JV., et al. Blood. 2017;130:567-580), the functions of *Snora75* remain to be clarified. Furthermore, we conducted qPCR and evaluated the expression levels of previously identified lncRNAs, which play pivotal roles in cardiac remodeling with pressure overload such as *Mhrt*, *Chast*, *Chaer*, and *CHRF*. As shown in Supplementary Fig. 10c, there were no significant differences in these lncRNA levels between control and *Lionheart*-KO mice. We added comments about these points on page 7, paragraph 2, lines 4–14 in Results section. Also, we inserted a paragraph on page 12, paragraph 2, lines 9–19 in Discussion section.

Inserted sentences (on page 7, paragraph 2, lines 4-14):

To assess the global expression changes of other previously identified RNAs in *Lionheart*-KO mouse hearts after TAC surgery, we performed microarray analysis for the levels of primary-miRNAs (pri-miRNAs) and other lncRNAs. The levels of pri-miR-669a-3, pri-miR-709, pri-miR-1946b, and pri-miR-3962 were more than 2-times higher in *Lionheart*-KO mice than control mice (Supplementary Fig. 10a). Concerning the other lncRNAs, 5 lncRNAs were increased (fold change > 2) and 2 lncRNAs were decreased (fold change < 0.5) in *Lionheart*-KO mice compared to control mice (Supplementary Fig. 10b). Furthermore, we conducted qPCR and evaluated the expression levels of previously identified lncRNAs that play pivotal roles in cardiac remodeling with pressure overload such as *Mhrt*¹⁴, *Chast*¹⁵, *Chaer*¹⁶, and *CHRF*¹⁷. As shown in Supplementary Fig. 10c, there were no significant differences in these lncRNA levels between control and *Lionheart*-KO mice.

Inserted paragraph (on page 12, paragraph 2, lines 9-19):

To assess if *Lionheart* knockout in mice leads to deregulation of miRNAs and other lncRNAs, we carried out microarray analysis and demonstrated that several pri-miRNAs and lncRNAs are deregulated in *Lionheart*-KO mouse hearts (Supplementary Fig. 10. a and b). Among the upregulated miRNAs, it is reported that long-term miR-669a overexpression with AAV9 ameliorates cardiac systolic dysfunction in a dystrophic mouse model²³. Thus, we are speculating that the miR-669a upregulation in *Lionheart*-KO mouse hearts is a compensatory machinery against cardiac systolic dysfunction observed in *Lionheart*-KO mice with pressure overload. Among the deregulated 7 lncRNAs, a paper showed that *Snora75*, a noncoding small nucleolar RNA, is negatively regulated by miR-24 which is enriched in platelet-derived microparticles (PMPs). While the PMPs treatment led to tumor cell apoptosis resulting in inhibition of cancer cell growth *in vivo* and *in vitro*²⁴, the functions of *Snora75* remain to be clarified and further investigations are needed.

2. In addition, the authors have not provided any direct information about, or insights into, the molecular/cellular/systemic mechanisms involved in mediating the functional effects of *Lionheart*. For example, it is not clear how pressure overload affects PURA expression and, in turn, how this affects *Lionheart*/LIONHEART expression?

Thank you for this important suggestion. We assessed PURA protein levels by western blotting to examine if pressure overload affects PURA expression in control and *Lionheart*-KO mouse hearts. As shown in Supplementary Fig. 12, there was no difference in PURA levels. Thus, it is unlikely that PURA protein levels affect the *Lionheart*/PURA interaction after pressure overload. We added these data as Supplementary Fig. 12 and changed a sentence about this point on page 8, paragraph 2, lines

10–11 in our revised manuscript.

Modified sentence (on page 8, paragraph 2, lines 8-11):

RNA immunoprecipitation using heart nuclear proteins and an anti-PURA antibody also demonstrated that PURA bound to *Lionheart* in the nucleus and the binding was increased after TAC (Fig. 4c), whereas pressure overload did not affect the PURA protein levels (Supplementary Fig. 12).

3. It is unclear if the observed changes are cell/region specific and/or if there are systemic changes in PURA and *Lionheart*/LIONHEART. Are there changes in these proteins/transcripts in the right ventricles following pressure overload induced cardiac hypertrophy?

In accordance with the reviewer's comment, we assessed *Lionheart* and PURA levels in right ventricles following pressure overload. As shown in Supplementary Fig. 13, there were no differences in *Lionheart* and PURA expression levels in the right ventricles with TAC surgery. These data indicate that TAC surgery resulted in left ventricle-specific *Lionheart* upregulation. We added comments about this point on page 8, paragraph 2, lines 11–17 in our revised manuscript.

Inserted sentences (on page 8, paragraph 2, lines 11-17):

To determine if the observed *Lionheart* upregulation is a left ventricle-specific change in hearts with TAC or not, we evaluated *Lionheart* levels in right ventricles of the heart after TAC and found that *Lionheart* upregulation was not observed in right ventricles after TAC (Supplementary Fig. 13a). Western blotting for PURA also demonstrated that PURA levels did not change in right ventricles after TAC (Supplementary Fig. 13b). These data indicate that TAC surgery resulted in left ventricle-specific *Lionheart* upregulation.

4. Also, it is unclear how *Lionheart* is linked to the various effects of pressure overload, including cellular hypertrophy and electrical remodeling, and tissue fibrosis, as these questions were not directly addressed in the studies reported in this manuscript. Much more could be done to explore/define the underlying mechanisms to increase the information/insights provided and to increase the impact of the work.

Thank you very much for this comment. In terms of cellular hypertrophy, we assessed the cross-sectional area of cardiomyocytes in control and *Lionheart*-KO mice following sham or TAC surgery and evaluated cardiomyocyte hypertrophy response to pressure overload. As shown in Figure A below, there was no difference in cardiomyocyte hypertrophy responses to pressure overload

between control and *Lionheart*-KO hearts. Based on these data, it is unlikely *Lionheart* is involved in cellular hypertrophy responding to pressure overload *in vivo*. In terms of tissue fibrosis and electrical remodeling, while the *Lionheart* level was higher in cardiomyocytes than in cardiac fibroblasts, *Lionheart* was detectable in cardiac fibroblasts as well (Fig. 2. b and d). Furthermore, as shown in Supplementary Fig. 7. g–j, cardiac fibrosis was significantly attenuated in *Lionheart*-KO hearts after TAC compared with control hearts. These data suggest that *Lionheart* in cardiac fibroblasts might be functional. Hence, we further carried out immunohistochemistry for proliferating cell nuclear antigen (PCNA) and found that interstitial cell proliferation was suppressed in *Lionheart*-KO mouse hearts compared with control mouse hearts as shown in Figure B below. Together, *Lionheart* in cardiac fibroblasts/myofibroblasts might regulate cell proliferation and the difference in cardiac fibrosis might affect electrical remodeling after pressure overload. However, we especially focused on the role of *Lionheart* in cardiomyocytes via sequestering PURA which regulates MYH6 expression in this manuscript. The *Lionheart* functions in cardiac fibroblasts/myofibroblasts might be uncovered in future years.

Figure A: Relative increase in cross-sectional area in control and *Lionheart*-KO mice after TAC.

Figure B: PCNA staining using control mice and *Lionheart*-KO mice at baseline. Blue: DAPI, Green: cardiac TnI, Red: PCNA

Response to Reviewer #2

We are so grateful to Reviewer #2 for the informative and useful comments. As described below, we have considered all of these comments and used them to improve our manuscript.

1. Phenylephrine, isoproterenol, and angiotensin II are regulators acting not only to cardiomyocytes, but also to blood pressure. How is the blood pressure, as a basic information of physiology of lionheart KO mice?

Thank you for this insightful suggestion. In accordance with this suggestion, we measured blood pressure in control and *Lionheart*-KO mice. As shown in Supplementary Fig. 8, there was no difference in blood pressure between the two groups. We added comments about this point on page 6, paragraph 1, lines 9–11 in our revised manuscript.

Inserted sentences (on page 6, paragraph 1, lines 9-11):

Because blood pressure affects cardiac hypertrophy, we measured blood pressure in control and *Lionheart*-KO mice. However, there was no difference in blood pressure between the two groups (Supplementary Fig. 8).

2. It is of interest that lionheart binds to PURA with highest specificity in hearts. As the authors mention, lionheart may a regulator of PURA to reduce the binding affinity to PNR. Thus, in lionheart KO mouse heart, PURA is able to bind to PNR with higher affinity than control heart, resulting in the repression of target genes, for example, Myh6. Is lionheart only one lincRNA that binds to PURA? If not, is there any RNA binding to PURA in lionheart KO mice, because of the lack of lionheart that occupies PURA with the highest affinity?

Thank you so much for this intriguing comment. As the reviewer imagines, we also assume that PURA has several binding partners and binding levels between PURA and other lincRNAs may change especially in *Lionheart*-KO mice with TAC surgery. To examine this, we conducted RNA immunoprecipitation using PURA antibody and assessed whether bindings between PURA and other PURA-binding lincRNAs are changed or not in *Lionheart*-KO mice. In terms of the other PURA-binding lincRNAs, we focused on *AV585709*, *Platr14*, *Platr27*, *Haunt*, and *Lockd* because Liu et al. previously reported that PURA binds to these five lincRNAs (Liu L., et al. *Nucleic Acids Res.* 2019;47:2244-2262). As shown in Figure A below, there were no differences in *AV585709*, *Platr14*, and *Platr27* between control and *Lionheart*-KO mice. The binding between PURA and *Haunt* tended to be increased in *Lionheart*-KO mice compared with controls ($p = 0.21$). In *Locked*, the binding was

significantly increased in *Lionheart*-KO mice compared with controls. These data indicate that the bindings between PURA and several other lncRNAs are increased in *Lionheart*-KO mice with TAC. However, based on the qPCR data in Figure B below, the expression levels of these lncRNAs in the heart are quite low compared with *Lionheart* and it's less than 2%. Also, the lncRNA levels were not increased in *Lionheart*-KO mice compared with controls under pressure overload (Figure B). Considering these results, we believe that these lncRNAs have little effect on the observed phenotype in *Lionheart*-KO mice.

Figure A: The interactions between PURA and PURA-binding lncRNAs including *AV585709*, *Platr14*, *Platr27*, *Haunt* and *Lockd*. n=4-6.

Figure B: The expression levels of *AV585709*, *Platr14*, *Platr27*, *Haunt*, and *Lockd* by qPCR. n=6–12.

3. Although the authors hypothesize that lionheart binds to PURA, it may be increased in lionheart KO hearts as a negative feedback, because it lacks the binding partner with the high affinity. Did the authors check the PURA level, for example, by Western blotting?

Thank you for this important suggestion. We assessed PURA protein levels by western blotting to examine if pressure overload affects PURA expression in control and *Lionheart*-KO mouse hearts. As shown in Supplementary Fig. 12, there was no difference in PURA levels. Thus, it is unlikely that PURA protein levels affect the *Lionheart*/PURA interaction after pressure overload. We added these data as Supplementary Fig. 12 and changed a sentence about this point on page 8, paragraph 2, lines 10–11 in our revised manuscript.

Modified sentence (on page 8, paragraph 2, lines 10-11):

RNA immunoprecipitation using heart nuclear proteins and an anti-PURA antibody also demonstrated that PURA bound to *Lionheart* in the nucleus and the binding was increased after TAC (Fig. 4c), whereas pressure overload did not affect the PURA protein levels (Supplementary Fig. 12).

4. It is well known that mitochondria play important roles in cardiac functions. The authors mention in p. 7, l. 3~12, lionheart may be involved in mitochondria. Are there any structural or biochemical changes in mitochondria of KO mice with/without TAC?

Thank you very much for this comment. In accordance with this suggestion, we observed mitochondrial structure by transmission electron microscopy. There was no obvious structural change between control and *Lionheart*-KO mouse hearts with or without TAC. In addition, we quantified mitochondrial size. There was no difference in mitochondrial size between control and *Lionheart*-KO mice. We added these data in Supplemental Fig. 11, and our explanation about these points on page 8, paragraph 1, lines 1–4.

Inserted sentences (on page 8, paragraph 1, lines 1-4):

Hence, we evaluated mitochondrial structure by transmission electron microscopy. However, there was no obvious difference in mitochondrial structures and sizes between control and *Lionheart*-KO mouse hearts (Supplementary Fig. 11).

Minor concerns:

In p. 10, l. 7, driven via may be better than driven by because of the redundant usage of by.

We fixed this sentence as shown below.

Lionheart expression is driven via its own promoter activated by a transcription factor, SRF.

REVIEWERS' COMMENTS:

Reviewer #1 (Remarks to the Author):

The authors have addressed my previous concerns in detail. I only have one minor comment and that is that I think that it would be better to state that specific experiments were designed/undertaken to "test" (in place of "prove") a hypothesis as presently stated (line 203) in the manuscript.

Reviewer #2 (Remarks to the Author):

Kuwabara et al. found upregulated lincRNAs obtained from TAC-treated mouse hearts and identified long intergenic-origin noncoding RNA in heart (lionheart). It is increased not only in mouse TAC model, but also angiotensin II- or phenylephrine-induced hypertrophied cardiomyocytes, is annotated at mm10_chr.2:77314725-77317068 on the minus strand, and is regulated transcriptionally by SRF via the promoter of the lionheart gene. The authors generated knockout (KO) mice, and observed structural and functional differences when treated with TAC after 8 weeks in KO hearts, in which Myh6, but not Myh7, was transcriptionally and translationally reduced. The authors tried to identify lionheart-binding proteins in mouse nuclear extracts of TAC-treated hearts, and of 46 candidates, due to the highest specificity to lionheart, they focused on Purine-rich element-binding protein A (PURA), a negative regulator of Myh6 transcription by binding to a purine-rich negative regulatory (PNR) element in the first intron of the Myh6 gene. They also conducted the rescue experiments using AAV9-lionheart with lionheart KO mice treated by TAC. Finally, the authors measured the lionheart levels in human biopsy specimens and showed positive and negative correlations with reduced cardiac systolic functions.

In the revised version, the authors addressed all concerns from this reviewer, and appropriately replied, added the new results according to the reviewer's comments, and rewrote the corresponding part of sentences of the results. The manuscript is greatly improved and includes new findings, which are very informative to broader readers not only in the medical and clinical, but also in the basic biological fields.

RESPONSES TO REVIEWERS' COMMENTS

We are grateful to Reviewer #1 and #2 for the important comments on our revised manuscript. As indicated in the following, we have taken all these comments and suggestions into account in the revised version of our manuscript.

Reviewer #1 (Remarks to the Author):

The authors have addressed my previous concerns in detail. I only have one minor comment and that is that I think that it would be better to state that specific experiments were designed/undertaken to "test" (in place of "prove") a hypothesis as presently stated (line 203) in the manuscript.

Thank you for your comment. According to this comment, we revised the sentence in the Main text as shown below.

Page 9, line 8: To test this hypothesis, we performed ChIP experiments using ...

Reviewer #2 (Remarks to the Author):

Kuwabara et al. found upregulated lincRNAs obtained from TAC-treated mouse hearts and identified long intergenic-origin noncoding RNA in heart (lionheart). It is increased not only in mouse TAC model, but also angiotensin II- or phenylephrine-induced hypertrophied cardiomyocytes, is annotated at mm10_chr.2:77314725-77317068 on the minus strand, and is regulated transcriptionally by SRF via the promoter of the lionheart gene. The authors generated knockout (KO) mice, and observed structural and functional differences when treated with TAC after 8 weeks in KO hearts, in which Myh6, but not Myh7, was transcriptionally and translationally reduced. The authors tried to identify lionheart-binding proteins in mouse nuclear extracts of TAC-treated hearts, and of 46 candidates, due to the highest specificity to lionheart, they focused on Purine-rich element-binding protein A (PURA), a negative regulator of Myh6 transcription by binding to a purine-rich negative regulatory (PNR) element in the first intron of the Myh6 gene. They also conducted the rescue

experiments using AAV9-lionheart with lionheart KO mice treated by TAC. Finally, the authors measured the lionheart levels in human biopsy specimens and showed positive and negative correlations with reduced cardiac systolic functions.

In the revised version, the authors addressed all concerns from this reviewer, and appropriately replied, added the new results according to the reviewer's comments, and rewrote the corresponding part of sentences of the results. The manuscript is greatly improved and includes new findings, which are very informative to broader readers not only in the medical and clinical, but also in the basic biological fields.

We appreciate the work of Reviewer #2 in reviewing our manuscripts. Thank you very much.